# Ultrathin ferrite nanosheets for room-temperature two-dimensional magnetic semiconductors

Ruiqing Cheng [1,5], Lei Yin[1,5], Yao Wen[1], Baoxing Zhai [1], Yuzheng Guo [2], Zhaofu Zhang[3], Weitu Liao[1], Wenqi Xiong[1], Hao Wang[1], Shengjun Yuan [1], Jian Jiang[1], Chuansheng Liu[1] & Jun He [1,4] ✉

The discovery of magnetism in ultrathin crystals opens up opportunities to explore new physics and to develop next-generation spintronic devices. Nevertheless, two-dimensional magnetic semiconductors with Curie temperatures higher than room temperature have rarely been reported. Ferrites with strongly correlated $d$-orbital electrons may be alternative candidates offering two-dimensional high-temperature magnetic ordering. This prospect is, however, hindered by their inherent three-dimensional bonded nature. Here, we develop a confined-van der Waals epitaxial approach to synthesizing air-stable semiconducting cobalt ferrite nanosheets with thickness down to one unit cell using a facile chemical vapor deposition process. The hard magnetic behavior and magnetic domain evolution are demonstrated by means of vibrating sample magnetometry, magnetic force microscopy and magneto-optical Kerr effect measurements, which shows high Curie temperature above 390 K and strong dimensionality effect. The addition of room-temperature magnetic semiconductors to two-dimensional material family provides possibilities for numerous novel applications in computing, sensing and information storage.

Two-dimensional (2D) materials, which cover almost all electronic states and functionalities in condensed matter, have attracted considerable interest due to their unique properties and competent performance that emerge in ultrathin thickness[1–5]. In particular, 2D magnetic order, which is associated with strong intrinsic spin fluctuations and had long been believed to hardly survive according to the Mermin–Wagner theorem, has been observed in atomically thin layers of $CrI_3$[6], $Cr_2Ge_2Te_6$[7], $FeTe$[8] and $CrSe_2$[9], opening the door to exploring new applications such as magnetic sensing and information storage. For the development of practical devices, 2D magnets must have intrinsic magnetic states that are stable at or above room temperature. Examples of recent success include $VSe_2$[10], iron oxides[11,12], and multiple chromium tellurides[13–15]. However, intrinsic room-temperature 2D magnetic semiconductors, which promises exciting technological opportunities for spin filtering and magnetic logic[5], has rarely been reported. Different from many conventional 2D materials whose properties are predominantly determined by the weakly interacting electrons in the $s$ and $p$ orbitals, complex transition metal oxides with strongly interacting $d$-orbital electrons host a wide range of correlated electronic phases such as high-temperature superconductivity and colossal magnetoresistance[16–20]. Recently, Ji et al. demonstrated the fabrication of one-unit-cell thick freestanding bismuth ferrite films by reactive molecular beam epitaxy (MBE), which exhibit giant tetragonality and polarization[16]. But, the antiferromagnetic order with

[1]Key Laboratory of Artificial Micro- and Nano-structures of Ministry of Education, and School of Physics and Technology, Wuhan University, Wuhan 430072, China. [2]School of Electrical Engineering and Automation, Wuhan University, Wuhan 430072, China. [3]The Institute of Technological Sciences, Wuhan University, Wuhan 430072, China. [4]Wuhan Institute of Quantum Technology, Wuhan 430206, China. [5]These authors contributed equally: Ruiqing Cheng, Lei Yin. ✉e-mail: He-jun@whu.edu.cn

nearly zero magnetization value, though higher than room temperature, hinders their realistic applications.

By contrast, the spinel (magnetite) family, another important class of complex oxides, usually exhibits relatively high magnetic ordering temperatures, large net magnetization, and stable chemistry property[21]. Besides, the intricate interplay of spin, charge, orbital, and lattice degrees of freedom brings them rich and complex physics, including Verwey transition[22], Jahn–Teller effect[23], spiral spin-liquid state[24], and magnetostriction[25–27]. Cobalt ferrite (CoFe$_2$O$_4$, CFO) is unique among the spinel ferrites in that it owns sizeable magnetic anisotropy due to a spin-orbit stabilized ground state[21,28,29]. Nevertheless, the synthesis of desirable non-van der Waals ultrathin complex oxides is still in its infancy because the inherent three-dimensional chemically bonded nature hinders their layer-by-layer exfoliation and 2D anisotropic growth. Recently, Kum et al. developed a remote epitaxy method of synthesizing CFO single-crystalline membranes with thickness of ~100 nm using pulsed laser deposition (PLD), providing a step towards 2D ferrites research[30]. But, restricted by the rigorous condition such as lattice-matched substrates and ultra-high vacuum, MBE and PLD method cannot be widely used in production. Besides, it will be interesting to explore whether the intrinsic magnetic order survives at a finite temperature in the 2D limit. Therefore, a facile and scalable approach (i.e., high throughput, low cost, transferability, and, importantly, compatibility with mature semiconductor technologies) to synthesize atomically thin spinel crystals is of significant importance for further development of the field.

In this work, we show that high-quality nonlayered CFO nanosheets as thin as a single unit cell can be synthesized via van der Waals epitaxy. The introduction of van der Waals substrates and confined conditions enables the van der Waals interactions between overlayers and substrates rather than the misfit energy to govern the crystal growth during the chemical vapor deposition (CVD) process. This strategy is also applicable to the synthesis of other spinel-type nanosheets. Electrical measurements reveals a typical semiconducting nature and a pronounced switchable effect for CFO nanosheets. The hard magnetic behavior and in situ magnetic domain evolution in CFO nanosheets are demonstrated by means of vibrating sample magnetometry (VSM), magnetic force microscopy (MFM), and magneto-optical Kerr effect (MOKE) measurements, which shows high Curie temperature above 390 K and strong dimensionality effect. Besides, we find no apparent change in magnetic properties after exposure to ambient conditions for over a month. We envision that the realization of room-temperature 2D magnetic semiconductors could greatly enrich the available 2D candidates and extend their application prospects.

## Results

### Van der Waals epitaxial growth of ultrathin CFO nanosheets

CFO typically crystallizes in a spinel-type nonlayered structure, in which two different metal ions (i.e., Co and Fe) occupy the centers of oxygen tetrahedrons and oxygen octahedrons, as schematically illustrated in Fig. 1a. The full cubic unit cell contains 56 atoms (i.e., eight formula unit cells) and its (111) lattice plane is highlighted by grey shadow. The closed-packed (111) planes have the lowest surface energy which makes the energetically favorable (111)-oriented spinel surfaces[31]. Top view of the projected plane along [111] zone axis shows the hexagonal arrangement of the atoms (Fig. 1a), and the side view reveals its nonlayered nature (Supplementary Fig. 1). The most attractive aspect of (111)-oriented spinel is the alternating triangular and Kagome magnetic atomic planes along the vertical direction, which is useful for studying the physics of correlated, frustrated and topological quantum states[32].

Figure 1b depicts the schematic illustration of van der Waals epitaxial growth of CFO nanosheets, which is carried out using a CVD system. Compared with MBE and PLD, CVD is most feasible for industrialization when one takes into account the high crystalline quality of 2D films as well as requirements of scalable-production and cost effectiveness. Freshly cleaved mica sheets with an atomically flat surface are positioned above the precursors and used as the growth substrates. Due to the small migration energy barrier of adatoms along the mica surface, the growth rate of the high-energy facets (i.e., the lateral direction) is much higher than that of the low-energy facets (i.e., the vertical direction)[33]. The narrow reaction space between the precursors and mica substrate could create a stable local environment and kinetically suppress the material growth along the vertical axis, hence enabling precise control of stoichiometric balance. In addition, the introduction of sodium chloride and molecular sieves, which has been proved to reduce the reaction temperature and facilitate the uniform evaporation of precursors[4,34], is essential for the epitaxial growth of ultrathin CFO nanosheets. Only disorderly stacked thick flakes are obtained without molecular sieves (Supplementary Fig. 2). More details on sample synthesis are provided in the Methods section. It is noteworthy that this strategy can be extended to the synthesis of other spinel-type nanosheets, such as manganese ferrites (Supplementary Fig. 3).

Figure 1c presents a typical optical microscope (OM) image of ultrathin CFO nanosheets with uniform optical contrast and identical crystallographic orientations, manifesting the epitaxial nature of CFO nanosheets on the mica surface. The seamless stitching of such unidirectionally aligned 2D islands has been proven to be effective in synthesizing wafer-scale 2D single crystals. They appear as regular triangle with the lateral size of tens of micrometers, indicating the sample inherited the hexagonal symmetry of CFO (111) lattice plane. The (111) preferred orientation of CFO nanosheets is further identified by X-ray diffraction (XRD) measurements (Supplementary Fig. 4). Statistics of AFM images demonstrate that the thickness of CFO nanosheets is mainly distributed in ~2–4 nm (Supplementary Fig. 5). Note that the van der Waals nature at the CFO/mica interface minimizes the substrate clamping effect and allows the easy release of CFO nanosheets for transferring onto other substrates or constructing artificial heterostructures, just like van der Waals layered materials. Thus we adopt a relatively mild etching-free method for the transfer of CFO nanosheets (Supplementary Fig. 6). The transferability of high-crystalline-quality ultrathin ferrites provides opportunities for novel interfacial physics and device applications.

Figure 1d shows the in-plane atomic-resolution high-angle annular dark-field scanning transmission electron microscopy (HAADF-STEM) image of CFO nanosheet, indicating a homogeneous structure. The spatial distribution of two different periodic atomic columns can be easily discriminated from the brightness contrast in HAADF-STEM image and the corresponding intensity line profile. Each bright atomic column is surrounded by six dark atomic columns arranged as hexagons, consistent with the atomic arrangement of the projected plane along [111] zone axis. Here, due to the complicated atoms distributing of spinel-type structure and the similar atomic number (Z) of Co/Fe, the brightness is related to the quantity of atoms rather than atomic number. Thus, the bright dots are atomic columns containing both tetrahedral sites and octahedral sites, while the dark dots are atomic columns containing only octahedral sites (Fig. 1e and Supplementary Fig. 1). As expected, the cross-sectional HAADF-STEM image of the sample shows a periodic rectangular pattern, consistent with the atomic arrangement of the projected plane along [1$\bar{2}$1] zone axis. The brightness contrast in cross-sectional image can be attributed to the fact that the quantity of octahedral sites per unit length in atomic column '1' is double of that in atomic column '2'.

Figure 1f shows the atomic force microscope (AFM) image of a typical CFO nanosheet, which features atomically smooth surface with root-mean-square roughness of ~0.1 nm and ultrathin thickness of ~1.7 nm corresponding to one unit cell along the [111] direction. The anisotropic ratio, which is described as the ratio of lateral size to

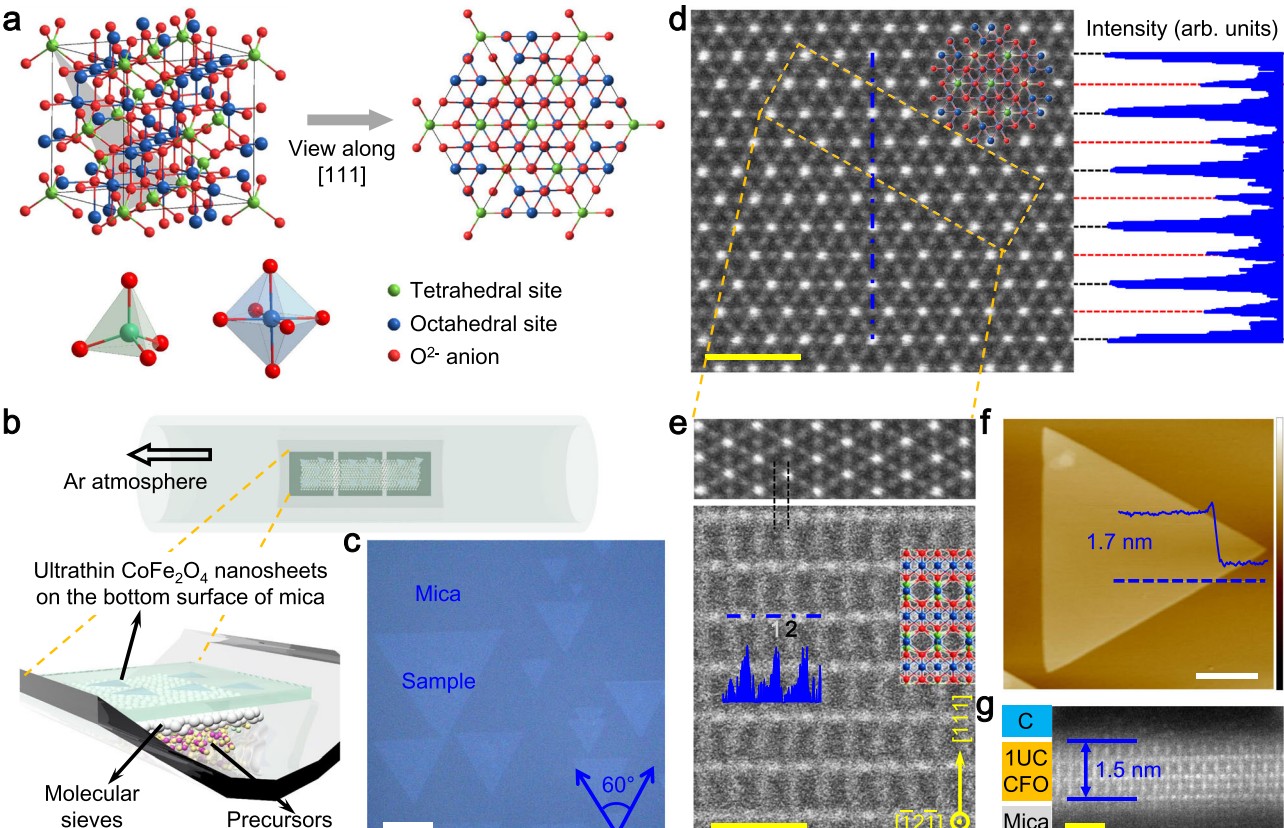

**Fig. 1 | Van der Waals epitaxial growth of ultrathin CFO nanosheets. a** The atomic model of CFO with spinel structure and its top view along the [111] zone axis. **b** Schematic illustration of epitaxial CFO nanosheets on mica substrate. **c** Optical microscopy image of the ultrathin triangular CFO nanosheets with a strong (111) orientation grown by van der Waals epitaxy. The image contrast is strengthened to improve its visibility. Scale bar: 10 μm. **d** Plan-view HAADF-STEM image of CFO nanosheet as well as the intensity profile taken along the dash-dot line, which is consistent with the atomic arrangement of the projected plane along the [111] zone axis. Scale bar: 1 nm. **e** Cross-sectional HAADF-STEM image of CFO nanosheet and the intensity profile taken along the dash-dot line, consistent with the atomic arrangement of the projected plane along the [$\bar{1}2\bar{1}$] zone axis. Two different atomic columns are labelled as '1' and '2', respectively. Top inset is the corresponding in-plan atomic arrangement. Scale bar: 1 nm. **f** AFM image of a typical CFO nanosheet. Inset is the corresponding height profile taken along the dash line. The thickness of ~1.7 nm corresponding to one unit cell along the [111] direction can be determined. Scale bar: 2 μm. **g** Cross-sectional HAADF-STEM image of 1-unit-cell CFO nanosheet, which shows an expanded out-of-plane lattice (~0.54 nm) in the bottom subcell. Scale bar: 1 nm.

vertical thickness, is estimated to be ~4800, much larger than the reported 2D metal oxides synthesized by wet-chemical methods[11,18]. The ultrathin thickness in the vertical direction could make the control of magnetic order through electrostatic gating, heterostructuring and strain more efficient compared with 1D or bulk architecture. We also perform cross-sectional studies on 1-unit-cell CFO nanosheet (Fig. 1g and Supplementary Fig. 7). Unexpectedly, the out-of-plane lattice of the bottom subcell expands to ~0.54 nm, whereas other subcells (~0.48 nm) are almost the same as that of the subcells observed in the thick CFO cross-section sample (Fig. 1e). This resembles the ultrathin freestanding bismuth ferrite films and may account for the lost magnetism of 1-unit-cell CFO[16].

**Sample characterizations and electrical properties**

High-resolution TEM (HRTEM) studies are applied to investigate the detailed information about the crystal structure of CFO nanosheets. Figure 2a–c shows the HRTEM image, selected area electron diffraction (SAED) pattern, and energy-dispersive X-ray spectroscopy (EDS) elemental mapping images of ultrathin CFO nanosheets, respectively. The interplanar distance between lattice fringes is measured to be 0.298 nm, corresponding to its (2$\bar{2}$0) planes. As shown, the HRTEM image and SAED pattern viewed along the [111] zone axis exhibit perfect hexagonally arranged lattice fringes and high-quality single-crystalline phase. In addition, the TEM-EDS elemental mappings confirm

that Co, Fe, and O are uniformly distributed throughout the entire crystal. Quantitative elemental analysis of CFO nanosheets is carried out through the corresponding TEM-EDS spectra, showing that the atomic percentage ratio of elements Co, Fe, and O is in good agreement with the stoichiometric ratio of $CoFe_2O_4$ (Supplementary Fig. 8). These results suggest the high crystal quality of the as-grown CFO nanosheets.

Raman spectra of CFO nanosheets excited by 532 nm laser are displayed in Fig. 2d and Supplementary Fig. 9. To explicitly identify the signal meant for CFO, Raman spectra of CFO nanosheets on different substrates (i.e., mica and $SiO_2$/Si) and pure mica sheet are conducted. Four prominent Raman peaks of CFO are observed at 692.7, 620.2, 470.6, and 304.8 cm$^{-1}$, which correspond to its $A^1_{1g}$, $A^2_{1g}$, $T^1_{1g}$ and $E_g$ modes, respectively. The high-frequency modes ($A_{1g}$) are usually attributed to the motion of oxygen atom in the tetrahedrons, while the other low-frequency modes ($T_{1g}$ and $E_g$) are related to the octahedrons[35]. The structural symmetry is studied by polarization-resolved second-harmonic generation (SHG) microscopy at room temperature (Fig. 2e). The non-zero polarized SHG signal reveals the broken inversion symmetry in the crystal that may induce electric polarization[36].

Figure 2f shows the $I_{DS}$−$V_{DS}$ curves of an 11.5 nm-thick CFO lateral device measured at different temperatures from 300 K to 100 K with a 10 K step. The corresponding OM and AFM images are displayed in

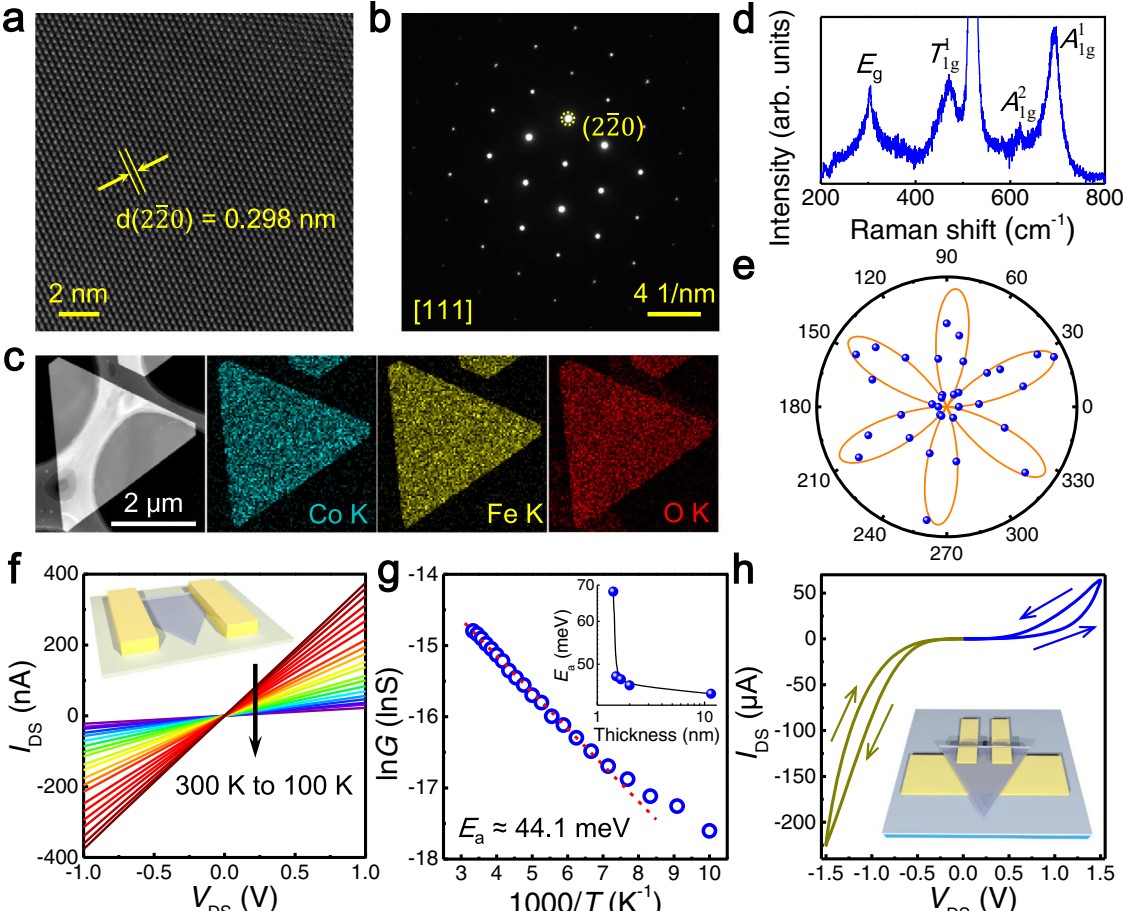

**Fig. 2 | Sample characterizations and electrical properties of CFO nanosheets.** HRTEM image (**a**), SAED pattern (**b**) and TEM-EDS elemental mapping images (**c**) of ultrathin CFO nanosheets. Here the lattice spacing is measured to be 0.298 nm, corresponding to its (2$\bar{2}$0) planes. **d** Raman spectrum of CFO nanosheet transferred onto silicon substrate. **e** Polarimetric SHG pattern obtained by rotating the CFO nanosheet while fixing the polarization of incident laser and collection analyzer. The solid line is the fitting curve. **f** $I_{DS}$−$V_{DS}$ curves of an 11.5 nm-thick CFO device measured at different temperatures from 300 K to 100 K with a 10 K step. Inset: The corresponding in-situ device schematic on mica substrate. **g** The Arrhenius plot of conductance ($G$) gives a thermal activation energy ($E_a$) of 44.1 meV. Inset: Plot of $E_a$ along with channel thickness. **h** $I_{DS}$−$V_{DS}$ curves of a 23 nm-thick CFO vertical device, showing switchable behavior. Sweep directions are indicated by arrows. Inset: Device schematic of CFO vertical device.

Supplementary Fig. 10. On the one hand, the linear $I_{DS}$−$V_{DS}$ characteristics indicate the device had good Ohmic contact within the test temperature range. On the other hand, the current decreases monotonously with the decreasing temperature, revealing the typical semiconducting nature of CFO nanosheets. The Arrhenius plot of conductance ($G$) shows very good linearity between 140 K and 300 K, confirming that the charge transport is dominated by thermal activation (Fig. 2g). The activation energy ($E_a$), defined as the energy difference between the Fermi level and mobility edge, is estimated to be 44.1 meV based on the thermally activation transport model $G(T) = G_0\exp(-E_a/k_BT)$; here, $G_0$ is the fitting parameter, $k_B$ is the Boltzmann constant and $T$ is the temperature[37,38]. The deviation from the linear fitting line at lower temperatures is derived from the charged impurities. In contrast, 1-unit-cell CFO nanosheets show relatively large $E_a$ values (the inset of Fig. 2g). We also fabricate top-gated CFO nanosheet devices through introducing high-$k$ dielectric, which exhibit a clear n-type transistor behavior (Supplementary Fig. 11).

Besides, a pronounced switchable effect is observed in CFO nanosheet vertical devices, as shown in Fig. 2h. The asymmetry between negative and positive bias conditions can be attributed to the different contacts at the two sides. Different states could be obtained after poling with opposite biases, suggesting the potential of CFO nanosheets for memory technology (Supplementary Fig. 12). Further, we investigate the electric switching behavior of CFO nanosheets

through conductive atomic force microscopy (CAFM), which also exhibits significant memory effect with switching ratio exceeding 10². Actually, the polar displacement of octahedral cations or native defects along the [111] direction has been reported for a number of spinel compounds, which gives rise to a dipole moment and be considered as the origin of ferroelectricity[39–41]. Hence, we suggest that the electric switching nature in CFO nanosheets may stem from the off-centre octahedrally coordinated ions with its polar axis normal to the 2D plane. The coupling between the mechanical and magnetic degrees of freedom may help to finds a variety of applications in magnetostrictive and magnetoelectric applications.

## Magnetic properties

We first calculate the electronic structure of CFO using the density functional theory (Supplementary Fig. 13). Although CFO exhibits the same macroscopic properties as ferromagnetic, namely a spontaneous and switchable net magnetization, its ground state magnetic structure is determined as ferrimagnetic (ferromagnetic Néel type). Figure 3a shows the theoretical models for bulk CFO and 1-unit-cell CFO, respectively. Bulk CFO is a magnetic semiconductor with a direct bandgap of 1.82 eV (Supplementary Fig. 14). The calculated density of states (DOS) gives semiconducting ground state and large spin-splitting for both bulk and 1-unit-cell models (Fig. 3b). The origin of magnetism is found to be the Fe 3$d$ orbitals near the bottom of the

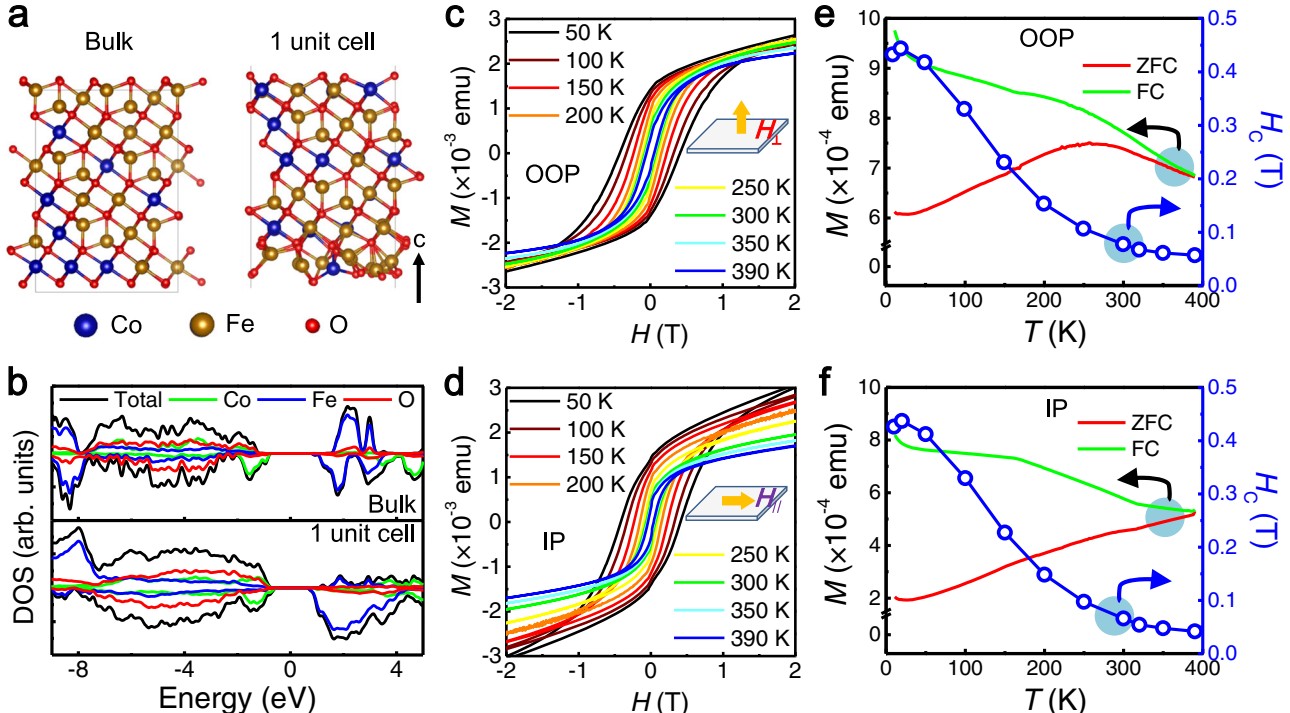

**Fig. 3 | Magnetic properties of CFO nanosheet samples.** Theoretical models (**a**) and calculated density of states (**b**) for bulk CFO and 1-unit-cell CFO, respectively. **c, d** Magnetic hysteresis loops of CFO nanosheet samples measured at different temperatures under out-of-plane (OOP) and in-plane (IP) magnetic field, respectively. **e, f** Temperature dependence of magnetic susceptibility ($M$, left axis, solid curves, measured with a field of 0.1 T) and coercive force ($H_C$, right axis, dotted curves) of CFO nanosheet samples under OOP and IP magnetic field, respectively.

conduction band and the Co 3$d$ orbitals in the upper part of the valence band.

Compared to bulk CFO, 1-unit-cell CFO possesses a smaller bandgap of 1.48 eV due to the reconstructed surface structure and the resulting local charge variation. The model is constructed by cutting of the O layer with a redistribution of O on the transitional metal layer. Besides, although the transition metal $d$ orbital nature does not change compared with the bulk case, we find the magnetic configuration near the surface deviates from the Néel-type spin alignment, and spin polarization is enhanced. This can be attributed to the lack of atoms in the other side of surface, which would be a major force directing the spins of transition metal atoms and exchange integrals. We also investigate its magnetic anisotropic energy (MAE), which is defined as the total energy difference between two different magnetization orientations, by considering the spin-orbit coupling. The results show that CFO exhibits a magnetic easy axis along the [111] direction. The magnetic anisotropy could resist the thermal fluctuations and lift the Mermin–Wagner restriction, and thus stabilize the long-range magnetic order in 2D regime.

The magnetic properties of CFO nanosheet samples with both out-of-plane (OOP, $H \perp$ CFO (111) plane) and in-plane (IP, $H$//CFO (111) plane) configurations are firstly investigated by susceptibility measurements, as shown in Fig. 3c–f and Supplementary Fig. 15. It is worth noting that the VSM measurement is carried out on an ensemble of CFO nanosheets collected from dozens of mica substrate, thus the average signal is obtained. Prominent open hysteresis loops are observed in $M$-$H$ curves under both OOP and IP magnetic field, indicating the long-range ferrimagnetic order in CFO nanosheets. Notably, the samples still exhibit well-defined hysteresis loops at 390 K, indicating the robust magnetism in CFO nanosheets and its potential for commercial (273 to 343 K) and industrial applications (233 to 358 K). The corresponding coercive force ($H_C$) decreases monotonically from 0.444 and 0.437 T at 20 K to 0.057 and 0.042 T at 390 K for OOP and IP configurations, respectively (Fig. 3e, f). The strong temperature

dependence of $H_C$ also indicates the ferrimagnetism of CFO nanosheets is not induced by defects[42]. Moreover, the rather large $H_C$ is indicative of a hard magnetic phase. Although CFO with (111) growth orientation exhibits a relatively weak magnetic anisotropy than (100), the comparison between $M$-$H$ curves under OOP and IP magnetic field at room temperature evidences an OOP magnetic easy axis, in accord with the calculated MAE and previous report[43].

For temperature-dependent magnetization ($M$-$T$) measurements, two modes, i.e., zero field cooling (ZFC) and field cooling (FC), are carried out in the temperature range of 10 to 390 K with a small field of 0.1 T. The FC magnetization shows a general trend of increasing with the decrease of temperature and diverge significantly from ZFC magnetization, demonstrating its ferromagnetic nature[15]. The magnetization is still observable at 390 K, matching well with the $M$-$H$ curves. The lift at low-temperature region (10 to 40 K) can be attributed to the mica substrate, which exhibits paramagnetism at low temperatures and diamagnetism at high temperatures (Supplementary Fig. 16). Besides, VSM results demonstrate that manganese ferrite nanosheets is a room-temperature soft magnetic material (Supplementary Fig. 17).

For imaging the magnetic structure of the as-grown samples, MFM measurements are conducted directly on the CFO nanosheets without any magnetization process. MFM phase images of CFO nanosheets with variable thicknesses are shown in Fig. 4a and Supplementary Fig. 18. The magnetostatic interaction between a magnetic tip and the stray micromagnetic fields from the sample causes the phase shift and thus creates the magnetic phase contrast, suggesting their room-temperature magnetism. With decreasing the nanosheet thickness, the magnetic structure shows a transition from multiple domain to single domain. Generally, domain structure is the result of exchange energy, anisotropy energy, and magnetostatic energy.

For the relatively thick nanosheet, they are partitioned into multiple uniform domains with anti-parallel magnetization to drastically reduce the magnetostatic energy, although the formation of domain walls increases the exchange energy and anisotropy energy to

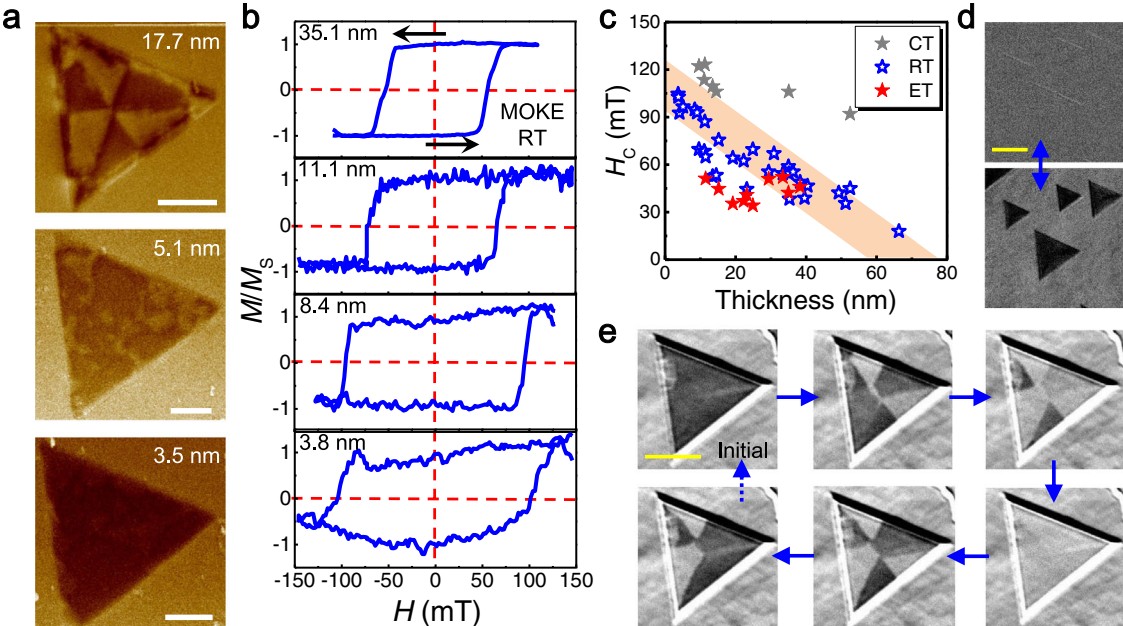

**Fig. 4 | Thickness-dependent magnetic properties of CFO nanosheets. a** MFM phase images of CFO nanosheets with variable thicknesses measured at room temperature (RT, 300 K). Scale bars: 3 μm. **b** MOKE hysteresis loops of CFO nanosheets with variable thicknesses measured at RT under OOP magnetic field. Sweep directions are indicated by arrows. **c** The plot of $H_C$ versus the thickness of CFO nanosheets at cryogenic temperature (CT, 80 K), RT, and elevated temperature (ET, 360 K), respectively. The shaded area is the guide of eyes to indicate the variation trend of $H_C$. MOKE microscopy images of typical thin (**d**) and thick (**e**) CFO nanosheets during the magnetization reversal at room temperature. The magnetization flip is uniform for the thin nanosheets. While for the thicker nanosheet, the magnetization start flipping in domains. Scale bars: 20 μm.

a certain degree within the transition region. It is noteworthy that their macroscopic magnetic arrangement is relevant to the real-space symmetry of crystal structure. As the sample thickness decreases, the reduction in the magnetostatic energy cannot compensate for the increase in the energy of domain walls, thus the nanosheets are in the energetically favorable vortex magnetic state. Their response to an external field is regarded as isotropic[44]. The corresponding critical thickness is determined as ~15 nm. When the thickness is reduced to about 2 unit cells (~3 nm), the magnetic configuration becomes a single domain, whose shape is exactly the same as the CFO nanosheet. Unfortunately, MFM test fails to demonstrate the ferrimagnetism of 1-unit-cell CFO. In contrast, the magnetic signal obtained from inner triangle island (~2 unit cells) is much stronger (Supplementary Fig. 18b).

The hard magnetic properties are further confirmed using MOKE microscope, which is considered as an ideal non-destructive optical method for in situ discerning 2D magnetism. Figure 4b shows the MOKE measurement of CFO nanosheets with variable thicknesses as a function of applied OOP magnetic field at room temperature. The MOKE signals consistently show typical hysteresis loops between the different field sweep directions and obvious remanence at zero magnetic field, manifesting a robust room-temperature ferrimagnetic order. Note that the observed magnetic phenomena are not due to the substrate effect by reason that (a) the interaction between overlayers and substrate are weak van der Waals forces and (b) CFO nanosheets on different substrates exhibit similar hysteresis loops (Supplementary Fig. 19).

When the CFO thickness reduced to ~2.5 nm, the MOKE signal is no longer a square hysteresis loop despite the obvious remanence (a sign of ferro-/ferrimagnetism). Further, no hysteresis loops are observed in sub-2-nm-thick CFO nanosheets. Even so, a sizeable Kerr rotation is observed as a function of the applied magnetic field, suggesting a detected OOP component of the magnetization. The reasons could be: (a) Enhanced spin fluctuation in reduced dimensions. Below critical dimensions, the formation of further domains becomes

energetically unfavorable, thus the magnetic moments flip randomly with time, and 1-unit-cell CFO behaves as super-paramagnetic by losing their magnetic order when the magnetic anisotropy energy becomes comparable to thermal energy. Such behavior is also observed in magnetized graphene and CFO nanoparticles[45,46]. (b) Due to the limited sensitivity of MFM and MOKE, it is difficult to determine exactly the weak stray magnetic field emanating from the 1-unit-cell CFO, just like many monolayer or 1-unit-cell cases in 2D crystals[7,14].

Figure 4c summarizes the trend of $H_C$ as a function of CFO thickness at cryogenic temperature (80 K), room temperature (300 K), and elevated temperature (360 K). Reliable reproducibility indicates the magnetic properties are less sensitive to the specific random magnetic cations distribution. The decrease of $H_C$ from 104.6 to 18.0 mT is observed with the increased nanosheet thickness from 3.7 to 66.3 nm at room temperature, revealing a strong dimensionality effect. The enhanced OOP anisotropy with decreasing thickness has already been reported in magnetic semiconductors such as $AgVP_2Se_6$ and $CrI_3$ nanosheets[47,48]. In addition, $H_C$ increases with the decreasing temperature due to the reduced thermal perturbation (Supplementary Figs. 20-21), consistent with the VSM studies. With this property, we can realize magneto-optical recording (i.e., record magnetic information by laser) more efficiently, since laser illumination could increase the local temperature and reduce the magnetic field required for magnetization reversal. Note that the MOKE signal shows little change after exposure to ambient conditions for over a month, demonstrating the well air stability of CFO nanosheets (Supplementary Fig. 22). Compared with the previously reported 2D magnets obtained by mechanical exfoliation of bulk materials (such as $CrI_3$ and $Cr_2Ge_2Te_6$)[6,7], the high Curie temperature, well environmental stability and high throughput make CFO nanosheets more practical in future spintronic devices.

Figure 4d shows the magnetization reversal images of typical thin samples with thicknesses of 9–14 nm captured using a MOKE microscope at room temperature. The uniform magnetization process indicates their response to external field is isotropic (i.e., coherent

magnetization rotation) and the entire crystal is substantially a magnetic single-domain during the reversal. The resulting high coercive forces promises to be of great practical importance in permanent magnet applications. In contrast, the fractional remanence, usually caused by randomly distributed micrometer-scale multi-domains, is observed in the relatively thick samples (~36.3 nm) during the reversal (Fig. 4e). In this case, the magnetization flip is determined by the domain wall displacement which usually requires relatively weak fields. Initially, the magnetization is saturated by applying a high magnetic field. As the field sweeping to a specific reverse value, the magnetization in certain regions switches direction. As the reverse field increases further, the remaining part starts to flip. Finally, the entire crystal completes spin-flip transition to align with the external field, and evolves into single-domain structure. When the field sweeping direction is flipped, the direction of magnetization reverses again. Such linear domain walls and magnetic domain pattern evolution are consistent with the results of MFM, and has also been observed in other 2D magnets such as $CrI_3$ and $Cr_2Ge_2Te_6$[6,7]. Our findings highlight the capacity of dimensionality toward the manipulation of magnetic domain evolution.

## Discussion

To summarize, we have developed a robust confined-van der Waals epitaxy strategy to synthesize air-stable CFO nanosheets with thickness down to one unit cell despite featuring out-of-plane covalent bonding, and realized a unique combination of semiconducting and room-temperature ferrimagnetic properties. VSM, MFM and MOKE measurements reveal that the resulting CFO nanosheets shows high Curie temperature above 390 K and strong dimensionality effect. We anticipate that the demonstration of intrinsic room-temperature magnetism in ultrathin semiconducting crystals offers opportunities for new quantum phenomena and device architectures, and gives a new impulse to the search for other 2D magnetic crystals that are absent in existing layered materials. For instance, we have demonstrated this synthetic strategy is also applicable to other spinel-type nanosheets. Moreover, ferrimagnets, which composed of antiferromagnetically coupled magnetic elements, can combine the advantages of both ferromagnets and antiferromagnets[49]. In this article, we have concentrated on its ferromagnetic properties, that is, a spontaneous and switchable net magnetization. Thus it is exciting to imagine the ultrafast magnetic moment dynamics brought by microscopic exchange interaction within the ferromagnetic Néel configuration and the potential for high-density devices.

As with any new material system, the lab-to-fab transitions require wafer-scale synthesis and scalable device fabrication. In our process, we show the potential of van der Waals epitaxy method in wafer-scale ultrathin ferrite film in future, and demonstrate that it is feasible to increase the ferrite coverage on the substrate by adjusting the growth parameters (Supplementary Fig. 23). To prepare wafer-scale ultrathin non-van der Waals film while maintaining uniformity and reliability, three conditions need to be satisfied: (1) the 2D anisotropic growth of nonlayered materials can be activated through modulating growth parameters; (2) the growth is implemented on a single-crystalline van der Waals substrate with proper surface symmetry and atomically flat surface; (3) seamless stitching of millions of unidirectionally aligned 2D islands to form large-area continuous films with well-controlled thickness and crystallographic orientation. We believe that the field of ultrathin non-van der Waals film growth at the wafer scale will continue to grow in importance in foreseeable future.

## Methods

### Sample synthesis and transfer
The synthesis was carried out using an atmospheric pressure chemical vapor deposition (APCVD) system equipped with 1-inch quartz tube. Weighed amount of ferric oxide ($\alpha$-Fe$_2$O$_3$, 99.5%, Aladdin), cobaltous oxide (CoO, 99.99%, Aladdin), ferric chloride (FeCl$_3$, 99.99%, Sigma Aldrich) and sodium chloride powders with the molar ratio of 1:1:0.2:0.01 were mixed evenly and used as precursors. A quartz boat with precursor powders and molecular sieves (3 A, 0.4-0.8 mm beads, Alfa Aesar) was placed in the center of the heating zone of the furnace, with fluorophlogopite mica substrates arrayed on the above. The quartz tube was purged with high-purity argon gas (Ar, 99.999%) at a flow rate of 500 sccm for 10 min. Then, the furnace was first heated from room temperature to 700 °C in 15 min and maintained at this temperature for another 15 min for sample growth. High-purity Ar (150 sccm) was used as the carrier gas to provide a suitable atmosphere. After the reaction completes, the furnace naturally cooled down to room temperature. Eventually, CFO nanosheets were deposited on the bottom surface of mica substrates. For the sample transfer, poly(methyl methacrylate) (PMMA) was spin-coated onto the sample (2000 rpm, 60 s) at first, followed by baking at 120 °C for 10 min. Then, the PMMA-capped CFO film was lifted off in the deionized water, followed by supporting the film with the required substrates (such as molybdenum grid and silicon substrate). The substrate with CFO film on the surface was dried at 100 °C for 10 min. Finally, hot acetone was used to remove the PMMA (60 °C for 10 min).

### Sample characterization
The morphology, thickness and material quality of CFO nanosheets were characterized by OM (Olympus BX51M), AFM (Bruker Dimension Icon) and Raman spectra (Horiba, 532 nm excitation laser) under the atmospheric environment, respectively. HAADF-STEM imaging measurements were taken at 300 kV on a FEI-Titan Cubed Themis G2 300 instrument equipped with double spherical aberration corrector. TEM imaging, SAED pattern and EDS measurements were performed on a JEM-F200. SHG measurement was conducted on a homemade second harmonic microscope equipped with a supercontinuum laser (Yslphotonics SC-PRO, 1300 nm excitation laser) under the atmospheric environment. The polarization of incident and SH lights was selected in parallel manner. The electrical properties were carried out on a probe station (Lakeshore TTP4) equipped with vacuum pump, flow cryostat, and Keithley 4200 semiconductor analyzer. All of the electrical measurements were performed under high vacuum (~$10^{-6}$ Torr) at room temperature unless otherwise specified. PFM measurements were carried out under ambient condition using contact mode AFM (Bruker Dimension Icon). A stiff tip with a spring constant of ~80 N/m (DDESP) was driven at 15 kHz. For CAFM measurements, the CAFM tip was placed on the surface of CFO nanosheet on Au coated SiO$_2$/Si substrate (Tip was kept grounded). A PtIr-coated Antimony doped Si tip with a tip radius of 25 nm and a spring constant of ~3 N/m (SCM-PIT-V2) was adopted. Magnetic measurements were performed by physical property measurement system (PPMS, Quantum Design) equipped with vibrating sample magnetometry utility, MFM (Bruker Dimension Icon, under interleave mode), and Kerr microscope with 617 nm wavelength LED (TuoTuo Technology, TTT-03).

### Device fabrication
For in-situ two-terminal CFO devices on the mica substrate, electrical contacts were first defined by mask, and followed by thermal deposition of 10/60 nm Cr/Au. For top-gate CFO devices, a 20 nm-thick hafnium oxide (HfO$_2$) dielectric layer was deposited by atomic layer deposition (ALD) on the prepared two-terminal devices. Subsequently, the top-gate electrode is fabricated by standard electron beam lithography (EBL) and metal deposition of 10/60 nm Cr/Au. For vertical devices, the bottom electrode is fabricated on SiO$_2$/Si substrate by standard EBL and metal deposition of 10/30 nm Cr/Au. Subsequently, CFO nanosheets were directionally transferred onto the pre-prepared bottom electrode, and followed by top electrode fabrication using the same method as used for the bottom electrode, forming an Au/CFO/Cr asymmetric vertical device structure.

## Theory calculations

First-principles calculations were implemented in the Vienna Ab-initio simulation package (VASP) based on density functional theory (DFT)[50]. The exchange correlation potential is described by Heyd-Scuseria-Ernzerhof (HSE) style hybrid functional[51]. The electron-ion potential is described by the projected augmented wave (PAW)[52]. The kinetic energy cutoff of plane wave is set to be 500 eV for the plane wave expansion. The Brillouin zone integration is carried out using $5 \times 5 \times 5$ Monkhorst-Pack $k$-point meshes for geometry optimization of CFO[53]. All geometric structures are fully relaxed until energy and forces are converged to $10^{-5}$ eV and 0.02 eV/Å, respectively. The climbing image nudged elastic band method[54] is used to determine the energy barrier of kinetic processes. The ground state magnetic structure is of ferromagnetic Néel type: that is, the magnetic moments of tetrahedral and octahedral cations are aligned in an antiparallel manner. In addition, the most equal/homogeneous cations distribution was adopted to preserve the highest possible symmetry and avoid possible system distortions, as suggested by Ansari et al.[55].

## Data availability

The data generated in this study are provided as a Source Data file. They are also available from the corresponding author upon reasonable request. Source data are provided with this paper.

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

## Acknowledgements

This work was supported by National Key R&D Program of China (No. 2018YFA0703700, J.H.), the National Natural Science Foundation of China (Nos. 91964203, J.H., 62104171, R.C., 62104172, L.Y., 62004142, Y.W., 62174122, Y.G. and 11774269, S.Y.), the Strategic Priority Research Program of Chinese Academy of Sciences (No. XDB44000000, J.H.), the Natural Science Foundation of Hubei Province, China (Nos. 2021CFB037, R.C. and 2020CFA041, S.Y.), and the Fundamental Research Funds for the Central Universities (No. 2042021kf0067, R.C.). The numerical calculations in this paper have been done on the supercomputing system in the Supercomputing Center of Wuhan University.

## Author contributions

J.H. supervised the project. R.C. and L.Y. conceived and designed the experiments. R.C. synthesized the sample and performed Kerr measurements. R.C. and L.Y. fabricated the devices and performed material characterization and electrical measurements. R.C. and Y.W. performed VSM measurements. B.Z., Y.G., Z.Z., W.L., W.X., and S.Y. carried out the theoretical calculation. R.C., L.Y., and J.H. analyzed the data and co-wrote the manuscript in consultation with all the other authors.

## Competing interests

The authors declare no competing interests.
