## [Peer Review File · Nature Communications]

Reviewers' Comments:

Reviewer #1:

Remarks to the Author:

The authors report on a method of synthesizing ultra-thin ferromagnetic CFO nanosheets, down to the 2-dimensional limit, on mica substrates via chemical vapor deposition. The key to achieving this remarkable result is using molecular sieves to ensure uniform evaporation of the precursors, a form of confined van der Waals epitaxial approach. Although this work is the first to report such thin CFO nanosheets, there are previous publications reporting growth of thin and flexible single-crystalline CFO membranes on mica substrates [Liu et al. ACS Appl. Mater. Interfaces 9, 7297 (2017)] using pulsed laser deposition. Nevertheless, fabrication of stable room temperature 2D ferromagnets using a more easily accessible CVD technique is of interest to the 2D material and solid-state device community.

The CFO flakes are also characterized meticulously, including structural, electrical, and magnetic properties along with band structure and density of states simulations, which match well with the experimental results. Their method is applicable to not only CFO but other spinel ferrites as well, demonstrated by synthesis of manganese ferrite nanosheets. It is my opinion that the work can be published in Nature Communications provided that the authors can revise the manuscript convincingly to address the following questions:

1. The biggest limitation of the current work seems like the area of the CFO nanosheets. One of the bigger issues in the 2D community is obtaining wafer-scale 2D materials that is uniform over the entire substrate. The biggest nanosheet shown in this work is at most a few tens of micrometers, which severely limit its applicability outside academic studies. Please add a comment on the scalability of this process such that it could go from "lab to fab".
2. How is the thickness uniformity of the CFO nanosheets over the entire substrate? Is the VSM measurement carried out on an ensemble of flakes or just one flake?
3. It is unfortunate that no magnetic characteristics could be measured for samples thinner than 2.7 nm, which is still thicker than a unit cell CFO. It seems the authors should have the tools and resources needed to clarify the reason, such as STM, HR-TEM etc.?
4. The manuscript needs to highlight the differences between ultrathin 2D CFO and bulk CFO (~50 nm), and how 2D CFO can expand the application space that bulk CFO cannot achieve.
5. Bulk CFO is known to be a ferrimagnet to the knowledge of this reviewer. Could the authors be more specific as to why they have decided to call it a ferromagnet in page 10, line 257?
6. What are the differences between ultrathin CFO and bulk CFO in terms of electronic band structure and density of states, and how does that affect the magnetic properties?
7. One of the more useful properties of CFO is its large magnetostriction. However, no magnetostriction characterization is shown in this work for the CFO nanosheets. It would be very useful for the community if the magnetostriction coefficient can be measured.
8. It seems that the CFO flakes exhibit both ferromagnetic and piezoelectric properties. Does this make it a multiferroic material? If it is, does simulation support this conclusion? I have some doubts about the piezoelectric measurement results and suspect spurious signals from the substrate, which often occurs using AFM measurement setups.
9. In Figure 4c, what does the peach-colored shaded area represent?
10. In Figure 4d,e, what is the critical thickness in which the CFO acts as a single domain magnet? Could the authors add an explanation as to why ultrathin CFO flakes flip magnetization like a single-domain whereas thicker (but not that much thicker I assume) CFO flakes start flipping in domains?

Response to Reviewer #1

The authors report on a method of synthesizing ultra-thin ferromagnetic CFO nanosheets, down to the 2-dimensional limit, on mica substrates via chemical vapor deposition. The key to achieving this remarkable result is using molecular sieves to ensure uniform evaporation of the precursors, a form of confined van der Waals epitaxial approach. Although this work is the first to report such thin CFO nanosheets, there are previous publications reporting growth of thin and flexible single-crystalline CFO membranes on mica substrates [Liu et al. ACS Appl. Mater. Interfaces 9, 7297 (2017)] using pulsed laser deposition. Nevertheless, fabrication of stable room temperature 2D ferromagnets using a more easily accessible CVD technique is of interest to the 2D material and solid-state device community.

The CFO flakes are also characterized meticulously, including structural, electrical, and magnetic properties along with band structure and density of states simulations, which match well with the experimental results. Their method is applicable to not only CFO but other spinel ferrites as well, demonstrated by synthesis of manganese ferrite nanosheets. It is my opinion that the work can be published in Nature Communications provided that the authors can revise the manuscript convincingly to address the following questions:

Our reply: First of all, we are grateful to the Reviewer for his/her positive evaluation on the manuscript and the fact that he/she thinks our work can be published in Nature Communications after revision.

We also thank the Reviewer for the instructive comments that stimulate us to further improve our manuscript. In the revised edition, we have addressed the Reviewer's questions and revised our manuscript accordingly. Our point-by-point responses are stated as below.

Comment 1: *The biggest limitation of the current work seems like the area of the CFO nanosheets. One of the bigger issues in the 2D community is obtaining wafer-scale 2D materials that is uniform over the entire substrate. The biggest nanosheet shown in this work is at most a few tens of micrometers, which severely limit its applicability outside*

academic studies. Please add a comment on the scalability of this process such that it could go from “lab to fab”.

Our reply: We agree with the Reviewer that the growth of wafer-scale single-crystal 2D ferrites is critically important for a variety of high-end applications. In our process, we have demonstrated the synthesized CFO nanosheets own almost identical crystallographic orientations, manifesting the epitaxial nature of CFO nanosheets (Fig. 1c and Supplementary Fig. S5b). The seamless stitching of millions of such unidirectionally aligned 2D islands has been proven to be effective in synthesizing wafer-scale 2D single crystals (*Nature* 2022, 605, 69–75; *Nat. Nanotechnol.* 2022, 17, 33–38; *Nat. Nanotechnol.* 2021, 16, 1201–1207).

Following your critical suggestion, we have tried to further optimize the growth parameters to increase the coverage area of CFO nanosheets. **Fig. R1** shows the optical microscope image and AFM image of CFO nanosheets with substrate temperature of 670 °C, which show higher coverage (up to 70%) but larger thickness (~14-18 nm) compared with the conditions described in the manuscript (700 °C). We have to admit that obtaining wafer-scale 2D ferrites with ultrathin thickness over the entire substrate is a really hard thing in a short timeframe. Nonetheless, we showed the potential of van der Waals epitaxy method in wafer-scale 2D ferrites, and we will conduct more experiments in the mentioned direction in future.

As advised by the Reviewer, we discussed the scalability of our process in the revised Manuscript as:

- (a) *“The seamless stitching of such unidirectionally aligned 2D islands has been proven to be effective in synthesizing wafer-scale 2D single crystals.”* (page 6, lines 1-2)
- (b) *“As with any new material system, the lab-to-fab transitions require wafer-scale synthesis and scalable device fabrication. In our process, we show the potential of van der Waals epitaxy method in wafer-scale ultrathin ferrite film in future, and demonstrate that it is feasible to increase the ferrite coverage on the substrate by adjusting the growth parameters (Supplementary Fig. 23). To prepare wafer-scale ultrathin non-van der Waals film while maintaining*

uniformity and reliability, three conditions need to be satisfied: (1) the 2D anisotropic growth of nonlayered materials can be activated through modulating growth parameters; (2) the growth is implemented on a single-crystalline van der Waals substrate with proper surface symmetry and atomically flat surface; (3) seamless stitching of millions of unidirectionally aligned 2D islands to form large-area continuous films with well-controlled thickness and crystallographic orientation. We believe that the field of ultrathin non-van der Waals film growth at the wafer scale will continue to grow in importance in foreseeable future.”
(page 16, lines 19-29)

Fig. R1 | Optical microscope image (a) and AFM image (b) of CFO nanosheets with substrate temperature of 670 °C.

Comment 2: *How is the thickness uniformity of the CFO nanosheets over the entire substrate? Is the VSM measurement carried out on an ensemble of flakes or just one flake?*

Our reply: We thank the Reviewer to point out this issue. Following your critical suggestion, we statistically analyzed the thickness and orientation of CFO nanosheets on a large scale. As shown in **Fig. R2**, statistics of AFM images demonstrate that the thickness of CFO nanosheets is mainly distributed in ~2-4 nm. Even though it is difficult to synthesize non-layered samples with uniform thickness at present, we obtained a relatively narrow thickness distribution of the resulting nanosheets.

As for the VSM measurement, it is carried out on an ensemble of CFO nanosheets

(the thickness is mainly distributed in $\sim 2\text{-}4$ nm). Thus the average signal is obtained when using VSM as the measurement method. To accurately determine the magnetic properties of a CFO nanosheet with well-defined thickness, we conducted MFM and MOKE measurements. These methods can be combined together and compensate each other, characterizing the magnetic properties of CFO nanosheets from different aspects.

Based on your comment, we added the statistical data, as well as the corresponding description to the revised manuscript and SI. Please see the marked text on page 6 (lines 6-7), page 11 (lines 23-25) and Supplementary Fig. 5.

Fig. R2 | Histogram statistics of the thickness (**a**) and orientation (**b**) of CFO nanosheets with substrate temperature of 700 °C, smooth curve is the Gaussian fit of the thickness distribution. The resulting nanosheets exhibit a relatively narrow thickness distribution ($\sim 2\text{-}4$ nm) and nearly identical crystallographic orientations.

Comment 3: *It is unfortunate that no magnetic characteristics could be measured for samples thinner than 2.7 nm, which is still thicker than a unit cell CFO. It seems the authors should have the tools and resources needed to clarify the reason, such as STM, HR-TEM etc.?*

Our reply: We agree with the Reviewer that exploring the magnetic properties of 1-unit-cell CFO is worthy of researching. In our previous version, we have demonstrated that 2-unit-cell CFO nanosheet is indeed a hard magnetic material. Following your critical suggestion, we carried out a certain amount of study on 1-unit-cell CFO cross-section sample and obtained clear cross-sectional HAADF-STEM atomic images. As shown in **Fig. R3**, the image shows high crystalline quality in 1-unit-cell-thick CFO

nanosheet, which owns a periodic rectangular pattern in the vertical direction, corresponding to three subcells with different atomic distributions. Unexpectedly, the out-of-plane lattice of the bottom subcell expands to ~ 0.54 nm, whereas the out-of-plane lattice of other subcells (~ 0.48 nm) are almost the same as that of the subcells observed in the thick CFO cross-section sample (Fig. 1e). This resembles the ultrathin freestanding BiFeO₃ films (*Nature* 2019, **570**, 87-90) and may account for the lost magnetism of 1-unit-cell CFO.

Fig. R3 | Cross-sectional HAADF-STEM image of 1-unit-cell CFO nanosheet. (a) SEM image of the 1-unit-cell CFO nanosheet sample prepared by Focused Ion Beam milling. (b,c) Cross-sectional HAADF-STEM image of 1-unit-cell CFO nanosheet, which shows a periodic rectangular pattern in the vertical direction, corresponding to three subcells with different atomic distributions.

Besides, more room-temperature MOKE and MFM data are provided (**Fig. R4**). When the CFO thickness reduced to ~ 2.5 nm, the MOKE signal is no longer a square hysteresis loop despite the obvious remanence at zero magnetic field (a sign of ferro-/ferrimagnetism). Further, no obvious hysteresis loops are observed in sub-2-nm-thick CFO nanosheets. Even so, a sizeable Kerr rotation is observed as a function of the applied magnetic field, suggesting a detected out-of-plane component of the

magnetization. These results are consistent with the MFM tests (**Fig. R4c**). MFM test fails to demonstrate the ferro-/ferrimagnetism of 1-unit-cell CFO nanosheet. In contrast, the magnetic signal obtained from inner triangle island (corresponding to ~2 unit cells) is much stronger.

By combining MFM and MOKE measurements, we can conclude that 2-unit-cell CFO is indeed a hard magnetic material. While for sub-2-nm-thick samples (~1 unit cell), we can only clarify that they own a detected out-of-plane component of the magnetization. The reasons could be:

- (1) Enhanced spin fluctuation in reduced dimensions. As compared to the bulk system, the decreased size of ferro-/ferrimagnetic material to a critical dimension may leads to single domain structure. Single domain structures exhibit large magnetic moment. Below critical dimensions, the formation of further domains becomes energetically unfavorable, thus the magnetic moments flip randomly with time and ferro-/ferrimagnetic material behaves as super-paramagnetic (SPM) by losing their magnetic order when the magnetic anisotropy energy becomes comparable to thermal energy. Such behavior is also observed in graphene on antiferromagnet CrSe (*Nat. Electron.* 2020, 3, 604–611, please see **Fig. R4b**), as well as CoFe₂O₄ nanoparticles with an average particle size of 15 nm (*Physica B* 2019, 567, 87-94).
- (2) Limited sensitivity of present instruments. Due to the limited sensitivity of MFM and MOKE, it is difficult to determine exactly the weak stray magnetic field emanating from the 1-unit-cell samples, just like many monolayer or 1-unit-cell cases in 2D atomic crystals (*Nature* 2017, 546, 265–269; *Nat. Commun.* 2021, 12, 5688). Recently, scanning magnetometry based on a single electron spin of a nitrogen-vacancy center in diamond has shown a high sensitivity in probing 2D magnets (*Science* 2021, 374, 1140–1144). Unfortunately, due to the limited condition in our lab and the influence of COVID-19, this characterization couldn't be conducted. In the future, we will conduct more experiments to validate the explanation.

Based on your comment, we have added the corresponding discussion into the

revised Manuscript as:

- (a) *“We also perform cross-sectional studies on 1-unit-cell CFO nanosheet (Fig. 1g and Supplementary Fig. 7). Unexpectedly, the out-of-plane lattice of the bottom subcell expands to ~ 0.54 nm, whereas other subcells (~ 0.48 nm) are almost the same as that of the subcells observed in the thick CFO cross-section sample (Fig. 1e). This resembles the ultrathin freestanding bismuth ferrite films and may account for the lost magnetism of 1-unit-cell CFO¹⁶.”* (page 7, lines 8-14)
- (b) *“MFM test fails to demonstrate the ferrimagnetism of 1-unit-cell CFO. In contrast, the magnetic signal obtained from inner triangle island (~ 2 unit cells) is much stronger (Supplementary Fig. 18b).”* (page 13, lines 21-23)
- (c) *“When the CFO thickness reduced to ~ 2.5 nm, the MOKE signal is no longer a square hysteresis loop despite the obvious remanence (a sign of ferro-/ferrimagnetism). Further, no hysteresis loops are observed in sub-2-nm-thick CFO nanosheets. Even so, a sizeable Kerr rotation is observed as a function of the applied magnetic field, suggesting a detected OOP component of the magnetization. The reasons could be: (a) Enhanced spin fluctuation in reduced dimensions. Below critical dimensions, the formation of further domains becomes energetically unfavorable, thus the magnetic moments flip randomly with time and 1-unit-cell CFO behaves as super-paramagnetic by losing their magnetic order when the magnetic anisotropy energy becomes comparable to thermal energy. Such behavior is also observed in magnetized graphene and CFO nanoparticles^{45,46}. (b) Due to the limited sensitivity of MFM and MOKE, it is difficult to determine exactly the weak stray magnetic field emanating from the 1-unit-cell CFO, just like many monolayer or 1-unit-cell cases in 2D crystals^{7, 14}.”* (page 14, lines 9-21)

Fig. R4 | (a) Room-temperature MOKE hysteresis loops of ultrathin CFO nanosheets with variable thicknesses. (b) The Figure extracted from *Nat. Electron.* 2020, 3, 604–611, which describes MOKE measurements of graphene on CrSe and a control sample. Kerr rotation of graphene on CrSe shows the magnetized graphene at 12 K and no magnetic signal at room temperature. (c) MFM phase images and AFM images of CFO nanosheets with variable thicknesses measured at room temperature. Scale bars: 3 μ m.

Comment 4: *The manuscript needs to highlight the differences between ultrathin 2D CFO and bulk CFO (~50 nm), and how 2D CFO can expand the application space that bulk CFO cannot achieve.*

Our reply: Thanks for bringing this issue to our attention. It is evident that the materials

with decreasing thickness has certainly resulted in unexpected properties since charge carriers, phonon and photon transports are strongly confined into 2D plane. Their 2D planar structures not only own high compatibility with flexible applications and traditional manufacturing process, but also provide a possibility of devising novel solid-state devices via stacking different layers (*Nature* 2022, 606, 902–908).

As for 2D CFO, on the one hand, the ultrathin thickness in the vertical direction could make the control of magnetic order through electrostatic gating, heterostructuring and strain more efficient compared with 1D or bulk architecture. On the other hand, the single-domain behavior (i.e. coherent magnetization rotation) and high coercive forces (~6 times larger than bulk CFO crystals) of ultrathin nanosheets promises to be of great practical importance in permanent magnet applications. While for relatively thick CFO, they are partitioned into multiple domains with anti-parallel magnetization. The multi-domain reversal is determined by the domain wall displacement which usually requires relatively weak fields. More discussion about the magnetic domain differences between ultrathin 2D CFO and bulk CFO is given in our **Reply 10**.

Based on your comment, we highlight the differences between ultrathin 2D CFO and bulk CFO. The corresponding discussion has been added to the revised manuscript. Please see the marked text on page 7 (lines 6-8), page 13 (lines 10-21) and page 15 (lines 10-19).

Besides, based on your **Comment 6**, we will discuss the differences between ultrathin CFO and bulk CFO in terms of electronic band structure, density of states and magnetic properties in our **Reply 6**.

Comment 5: *Bulk CFO is known to be a ferrimagnet to the knowledge of this reviewer. Could the authors be more specific as to why they have decided to call it a ferromagnet in page 10, line 257?*

Our reply: Many thanks for this kindly reminder. The reason why we call it a ferromagnet is that 2D CFO nanosheets exhibits the same macroscopic properties as ferromagnetic: namely, a spontaneous and switchable net magnetization. In reference to some literatures, ferrimagnetic (FiM) structure (the magnetic moments of tetrahedral

and octahedral cations are aligned in antiparallel manner) is also referred as ferromagnetic Néel type (*Phys. Rev. B* 2020, 102, 035446; *ACS Nano* 2017, 11, 8002–8009).

After carefully considering your comment and further discussion with my colleagues, we agree that ferrimagnet is a more scientific and orthodox expression. Accordingly, although the exhibited ferromagnetic macroscopic properties of CFO nanosheets are superior to most of 2D ferromagnets, we adopted “*ferrimagnetic*” instead of “*ferromagnetic Néel type*” in the revised manuscript to avoid possible misunderstanding. We also modified the corresponding text for more accurate description as “*Although CFO nanosheets exhibits the same macroscopic properties as ferromagnetic: namely, a spontaneous and switchable net magnetization, its ground state magnetic structure is determined as ferrimagnetic (ferromagnetic Néel type).*” (page 10, lines 9-11)

Besides, we highlighted the uniqueness of ferrimagnets in the revised Manuscript as “*Moreover, ferrimagnets, which composed of antiferromagnetically coupled magnetic elements, can combine the advantages of both ferromagnets and antiferromagnets (Nat. Mater. 2022, 21, 24–34). In this article, we have concentrated on its ferromagnetic properties, that is, a spontaneous and switchable net magnetization. Thus it is exciting to imagine the ultrafast magnetic moment dynamics brought by microscopic exchange interaction within the ferromagnetic Néel configuration and the potential for high-density devices.*” (page 16, lines 12-18)

Comment 6: *What are the differences between ultrathin CFO and bulk CFO in terms of electronic band structure and density of states, and how does that affect the magnetic properties?*

Our reply: Considering the nonlayered nature and large crystal cell (it containing 168 atoms per unit cell) of CFO crystal that requiring huge computing resource, we first calculated the electronic band for four kinds of CFO nanosheet structures with different exposed atoms at the surface based on LDA+U methods (**Fig. R5a**). The large differences between them may originate from different surface reconstruction and the resulting local charge variation (*Phy. Rev. Lett.* 2022, 128, 226102). However, all the

calculation results indicate that CFO nanosheet is a magnetic metal, which contradicts our experiment results (i.e. we demonstrated CFO nanosheets is a semiconductor, as shown in Fig. 2f, g, Supplementary Fig. 10 and 11). This indicates that the intrinsic properties of CFO is overshadowed by their complex surface states.

It is well known that the U values used in LDA+ U methods depend on the local environment (e.g. the coordination of transition metal atoms), especially for complex structure such as surface and interface. When we artificially change on-site U that is determined by Coulomb screening, the occupations of transition metal d orbitals change significantly especially in the 2D limit, resulting in the variation of DOS. Therefore in the revision, we adopted the more accurate but more expensive HSE hybrid functional for electronic structure, which gives semiconducting ground state and large spin-splitting for both bulk and 1-unit-cell models (**Fig. R5b,c**). The origin of magnetism is found to be the Fe $3d$ orbitals near the bottom of the conduction band and the Co $3d$ orbitals in the upper part of the valence band. Compared to bulk CFO with a bandgap (E_g) of 1.82 eV, 1-unit-cell CFO possesses a smaller bandgap of 1.48 eV due to the reconstructed surface structure and the resulting local charge variation. The model is constructed by cutting of the O layer with a redistribution of O on the transitional metal layer. Besides, although the transition metal d orbital nature does not change compared with the bulk case, we found the magnetic configuration near the surface deviates from the Néel-type spin alignment, and spin polarization is enhanced. This can be attributed to the lack of atoms in the other side of surface, which would be a major force directing the spins of transition metal atoms and exchange integrals.

Fig. R5 | **a**, Calculated electronic band for four kinds of CFO nanosheet structures with different exposed atoms at the surface. **b,c**, Theoretical models (**b**) and calculated density of states (**c**) for bulk CFO and 1-unit-cell CFO, respectively.

Based on your comment, we have added the corresponding discussion into the revised Manuscript as: “Figure 3a shows the theoretical models for bulk CFO and 1-unit-cell CFO, respectively. Bulk CFO is a magnetic semiconductor with a direct bandgap of 1.82 eV (Supplementary Fig. 14). The calculated density of states (DOS) gives semiconducting ground state and large spin-splitting for both bulk and 1-unit-cell models (Fig. 3b). The origin of magnetism is found to be the Fe 3d orbitals near the bottom of the conduction band and the Co 3d orbitals in the upper part of the valence band. Compared to bulk CFO, 1-unit-cell CFO possesses a smaller bandgap of 1.48 eV

due to the reconstructed surface structure and the resulting local charge variation. The model is constructed by cutting of the O layer with a redistribution of O on the transitional metal layer. Besides, although the transition metal d orbital nature does not change compared with the bulk case, we find the magnetic configuration near the surface deviates from the Néel-type spin alignment, and spin polarization is enhanced. This can be attributed to the lack of atoms in the other side of surface, which would be a major force directing the spins of transition metal atoms and exchange integrals.”

Please see the marked text on page 11 (lines 1-13).

Comment 7: *One of the more useful properties of CFO is its large magnetostriction. However, no magnetostriction characterization is shown in this work for the CFO nanosheets. It would be very useful for the community if the magnetostriction coefficient can be measured.*

Our reply: We agree with the Reviewer that one of the more useful properties of CFO is its large magnetostriction. CFO represents a unique case among all the ferrite families since its large magnetostriction can promote the strain anisotropy to prevail in 2D structures. The controllable deformation by a magnetic field is the most important feature of magnetostrictive effect. The usual magnetostriction measurement includes: (1) measuring the free end deflection of the cantilevered film-substrate system as a function of the external field; (2) using a digital holographic microscope to map out the magnetostriction deflection profile. Jen et al. and Liu et al. determined the magnetostrictive response of 130 nm-thick $\text{Fe}_{62}\text{Co}_{19}\text{Ga}_{19}$ films and 100 nm-thick CFO films, respectively (*IEEE Trans. Magn.* 2014, 50, 6000404; *ACS Appl. Mater. Interfaces* 2017, 9, 7297). As for CFO nanosheets, the ultrathin thickness (~2-4 nm) and relatively low coverage make it difficult for the detection of the deformation of CFO/mica bimorph under external field. As discussed in our **Reply 1**, in spite of our best efforts, obtaining wafer-scale 2D ferrites over the entire substrate is a really hard thing in a short timeframe.

It is also noteworthy that these measurements to estimate magnetostriction generally adopt the bulk Poisson's ratio, which might not apply to our CFO ultrathin nanosheets

(*ACS Appl. Mater. Interfaces* 2017, 9, 7297). We have tried to get results from MOKE microscope. However, the expected deformation degree of 25 nm is hard to distinguish by MOKE microscope. Here, we assumed that the lateral dimension of CFO nanosheet is 50 μm and the magnetostriction is 500 ppm (in parts per million, based on previous reports). Recently, Jiang et al. from Cornell University demonstrated an exchange-driven magnetostriction effect in mechanical resonators made of 2D CrI_3 , as the schematics shown in **Fig. R6** (*Nat. Mater.* 2020, 19, 1295–1299). The magnetostriction is estimated to be $\sim 1\text{--}10$ ppm, far below the $\sim 100\text{--}700$ ppm of CFO. However, such a complex mechanical device platform couldn't be built with the short period. We now feel it is a very valuable research field, and we will do more efforts in the mentioned direction in future.

Based on your comment, we highlighted the potential of 2D CFO nanosheets in magnetostrictive applications in the revised Manuscript as “*The coupling between the mechanical and magnetic degrees of freedom may help to finds a variety of applications in magnetostrictive and magnetoelectric applications*” (page 10, lines 5-7).

Fig. R6 | The Figure extracted from *Nat. Mater.* 2020, 19, 1295–1299, which describes the schematic of 2D CrI_3 mechanical resonators. The resonator (suspended 2D membrane on a Si trench of depth D) is actuated by an r.f. voltage from a vector network analyser (VNA) through a bias tee. A d.c. voltage V_g is superimposed to apply static

tension to the membrane. The motion is detected interferometrically by a HeNe laser, which is focused onto the centre of the resonator. BS: beam splitter; PD: photodetector.

Comment 8: *It seems that the CFO flakes exhibit both ferromagnetic and piezoelectric properties. Does this make it a multiferroic material? If it is, does simulation support this conclusion? I have some doubts about the piezoelectric measurement results and suspect spurious signals from the substrate, which often occurs using AFM measurement setups.*

Our reply: In experiment: (1) the non-zero polarized SHG signal reveals the broken inversion symmetry in the crystal that may induce electric polarization; (2) vertical device and conductive atomic force microscopy measurements indicate a pronounced out-of-plane switchable effect of CFO nanosheets, and different states could be obtained after poling with opposite biases; (3) piezoresponse force microscopy measurements indicate clear polarization switching of CFO flake-based metal–oxide–semiconductor structure. Nevertheless, as mentioned by the Reviewer, other contributions such as interface charging effect, substrate signals and species diffusion should not be excluded.

In theory, the polar displacement of octahedral cations or native defects along the [111] direction has been reported for a number of spinel compounds, which gives rise to a dipole moment and be considered as the origin of ferroelectricity (*Philos. Mag.* 1972, 26, 1217-1226; *Nature* 2005, 434, 364-367; *Phys. Rev. B* 2019, 99, 144412). Hence, we suggest that the electric switching nature in CFO nanosheets may stem from the off-centre octahedrally coordinated ions with its polar axis normal to the 2D plane.

By combining experiment and theory, we can conclude that CFO nanosheets owns broken inversion symmetry and electric switching nature. While for piezoelectricity, there is no conclusive evidence at current stage. We have clearly pointed out this in our previous version. In reference to the recently reported 2D multiferroic NiI₂ (*Nature* 2022, 602, 601–605), we suspect that CFO nanosheets may be an example of improper electronic ferroelectricity. Birefringence-induced polarization rotation measurement is probably the answer. Unfortunately, due to the limited condition in our lab and the

influence of COVID-19, this characterization couldn't be conducted with the short period. Based on your comment, we highlighted the broken inversion symmetry and electric switching nature of CFO nanosheets in the manuscript, and de-emphasized the ferroelectricity since this is not the main research object of this work: ultrathin ferrite nanosheets for room-temperature 2D magnetic semiconductors. We will do more efforts in the mentioned direction in future. The corresponding discussion has been added to Supplementary Fig. 12 in the revised Supplementary Information.

Comment 9: *In Figure 4c, what does the peach-colored shaded area represent?*

Our reply: The peach-colored shaded area in Figure 4c is only the guide of eyes to indicate the variation trend of coercive force (H_c). Based on your comment, the corresponding description has been added into the caption of Figure 4c (page 12) in the revised Manuscript as “*The shaded area is the guide of eyes to indicate the variation trend of H_c .*”

Comment 10: *In Figure 4d,e, what is the critical thickness in which the CFO acts as a single domain magnet? Could the authors add an explanation as to why ultrathin CFO flakes flip magnetization like a single-domain whereas thicker (but not that much thicker I assume) CFO flakes start flipping in domains?*

Our reply: The Reviewer raised a valid point. In our previous version, we showed that the magnetization flip of thin nanosheet (less than 14 nm) is uniform and the entire crystal is substantially a magnetic single-domain. While for the relatively thick CFO nanosheet, the magnetization start flipping in domains. These results are consistent with the MFM test. Thus, we using MFM with higher spatial resolution to determine the critical thickness in which the CFO nanosheet acts as a single domain magnet.

A magnetic domain is a region in which the magnetic spins of atoms or molecules are aligned. As shown in **Fig. R7**, with decreasing the nanosheet thickness, the magnetic structure shows a transition from multiple domain to single domain. Generally, domain structure is the result of exchange energy, anisotropy energy and magnetostatic energy.

(1) For the relatively thick nanosheets (**Fig. R7a, b**), they are partitioned into

multiple uniform domains with anti-parallel magnetization to drastically reduce the magnetostatic energy, although the formation of domain walls increases the exchange energy and anisotropy energy to a certain degree within the transition region (*Rev. Mod. Phys.* 1949, 21, 541). In this case, the magnetization flip is determined by the domain wall displacement which usually requires relatively weak fields. It is noteworthy that the corresponding magnetic configuration is relevant to the real-space symmetry of crystal structure.

(2) As the sample thickness decreases, the reduction in the magnetostatic energy cannot compensate for the increase in the energy of domain walls, thus the nanosheets are in the energetically favorable vortex magnetic state. Their response to an external field is regarded as isotropic (*Appl. Phys. Lett.* 2001, 78, 3848). Based on **Fig. R7b, c**, we determined the critical thickness in which the CFO nanosheet acts as a single domain magnet during the reversal is ~15 nm. When the thickness is reduced to about 2 unit cells, the magnetic configuration becomes a single domain and maintains its uniformity during the reversal.

(3) As the sample thickness further decreases to 1 unit cell, the magnetic signal is hard to detect. 1-unit-cell CFO may undergo the transition to super-paramagnetic by losing their magnetic order when the magnetic anisotropy energy becomes comparable to thermal energy. We have detailedly discussed this behavior in our **Reply 3**.

Based on your comment, we discussed why ultrathin CFO flakes flip magnetization like a single-domain whereas thicker CFO flakes start flipping in domains, and determined the critical thickness (~15 nm) in which the CFO nanosheet acts as a single domain magnet during the reversal. The corresponding characterization and discussion were added to the revised manuscript and SI. Please see the marked text on page 13 (lines 8-23), page 15 (lines 11-19) and Supplementary Fig. 18.

Fig. R7 | MFM phase images and AFM images of CFO nanosheets with variable thicknesses measured at room temperature. Scale bars: 3 μm.

Response to Reviewer #2

Manuscript reports the synthesis of room-temperature two-dimensional ferrite nanosheets. van der Waals epitaxial approach was used to synthesize cobalt ferrite nanosheets with thickness of one unit cell using a facile chemical vapor deposition process. Synthesized material exhibit high Curie temperature above 390 K and room-temperature ferromagnetic properties. Authors claimed that it is possible to use this material in numerous novel applications such as computing, sensing and information storage. Some of the results presented in the manuscript are very interesting and thus manuscript can be considered for publication subject to satisfactorily revision according to following comment;

Our reply: First of all, we are grateful to the Reviewer for his/her positive evaluation on the manuscript and the fact that he/she thinks our work can be considered for publication after revision.

We also thank the Reviewer for the instructive comments that stimulate us to further improve our manuscript. In the revised edition, we have addressed the reviewer's questions and revised our manuscript accordingly. Our point-by-point responses are stated as below.

Comment 1: *Convening structural analysis is missing in the manuscript. Reciprocal space mapping and XRD measurements should have been carried out to shed light on the orientation of the prepared material.*

Our reply: We thank the Reviewer to point out this issue. In our previous studies, the (111) orientation of the prepared CFO nanosheets is demonstrated by: (1) at single sheet level, TEM, plan-view and cross-sectional HAADF-STEM measurements indicate that CFO nanosheets is grown along the [111] direction (Fig. 1 and Fig. 2); (2) at substrate level, CFO nanosheets over the substrate appear as regular triangle (consistent with hexagonal symmetry of CFO (111) lattice plane) and exhibit identical crystallographic orientations, manifesting the epitaxial nature (Fig. 1c, Supplementary Fig. 5b, 23).

As advised by the Reviewer, we conducted additional XRD measurements to further identify the orientation of CFO nanosheets. As shown in **Fig. R8**, the XRD peaks

around 18.3° , 37.1° , 57.0° and 79.0° are corresponding to its (111), (222), (333) and (444) planes. The weak peak signals is due to the ultrathin nature of CFO nanosheets. Besides, no other peaks related to other planes are observed. These results indicate that our CFO nanosheets is not randomly oriented, and [111] is the preferred growth orientation.

Combining our previous studies and newly-added XRD characterization, we believe the crystal structure of CFO nanosheets is more clearly described in this revision. Based on your comment, we added the XRD data, as well as the corresponding description to the revised manuscript and SI. Please see the marked text on page 6 (lines 4-6) and Supplementary Fig. 4.

Fig. R8 | X-ray diffraction (XRD) spectra of CFO nanosheet samples on the mica substrate. The XRD peaks around 18.3° , 37.1° , 57.0° and 79.0° are corresponding to its (111), (222), (333) and (444) planes, confirming the epitaxial nature with (111) preferred orientation.

Comment 2: *Molecular beam epitaxy method on STEP single crystalline substrate can also grow 2D-monolayer of CFO. Thus, why authors choose the specific root mentioned in the manuscript.*

Our reply: It is true that ferrite ultrathin films can grow on a stepped single crystalline substrate by molecular beam epitaxy method, as illustrated in **Fig. R9**. The [001] axis

of the off-cut substrate is tilted by 6° towards the $[100]$ axis, thus, there are step edges along the $[010]$ axis on its surface. However, the additional epitaxial strain at the substrate/film interface, which depends on the angle and direction of the miscut, strongly affects the geometry and physical properties of ferrite (*Phys. Rev. B* 2013, 88, 195110; *Phys. Rev. B* 1998, 57, 3679). The strong interaction between the epitaxial layer and substrates also brings myriad fabrication challenges for further component processing. Besides, restricted by rigorous conditions such as lattice-matched substrates, ultra-high vacuum and slow growth rates, MBE method is not used in production.

Fig. R9 | The schematic illustration of molecular beam epitaxy of ferrite film on a stepped single crystalline substrate.

In view of the above, we choose the van der Waals epitaxy approach to synthesizing 2D ferrites using a facile and scalable chemical vapor deposition (CVD) process. On the one hand, the van der Waals nature at the ferrite/mica interface minimizes the substrate clamping effect and allows the detection of the intrinsic properties of ferrite nanosheets, as well as the easy release of ferrite nanosheets for transferring onto other substrates or constructing artificial heterostructures, just like van der Waals layered materials. On the other hand, compared with MBE and ALD, CVD is most feasible for industrialization when one takes into account the uniformity and crystallinity of 2D films as well as requirements of high throughput, cost effectiveness and scalability (*Nat.*

Commun. 2022, 13, 1484).

Based on your comment, we added the corresponding discussion to the revised manuscript as (i) “*But, restricted by the rigorous condition such as lattice-matched substrates and ultra-high vacuum, MBE and PLD method cannot be widely used in production. Besides, it will be interesting to explore whether the intrinsic magnetic order survives at a finite temperature in the 2D limit. Therefore, a facile and scalable approach (i.e. high throughput, low cost, transferability, and, importantly, compatibility with mature semiconductor technologies) to synthesize atomically thin spinel crystals is of significant importance for further development of the field*” (page 3, lines 5-12), (ii) “*Compared with MBE and PLD, CVD is most feasible for industrialization when one takes into account the high crystalline quality of 2D films as well as requirements of scalable-production and cost effectiveness*” (page 5, lines 11-14) and (iii) “*Note that the van der Waals nature at the CFO/mica interface minimizes the substrate clamping effect and allows the easy release of CFO nanosheets for transferring onto other substrates or constructing artificial heterostructures, just like van der Waals layered materials.*” (page 6, lines 7-10)

Comment 3: *Literature reports (e.g. ACS Appl. Electron. Mater. 2020, 2, 3650–3657) suggest that CFO exhibit magnetic easy axis along [100] direction. Please check it carefully. Need a reference for ‘CFO exhibits a magnetic easy axis along the [111] direction’.*

Our reply: In the literature report mentioned by the Reviewer, Shirsath et al. showed that CFO film with (100) orientation exhibit magnetic easy axis along [100] direction, while CFO film with (111) orientation exhibit its magnetic easy axis in the in-plane direction (*ACS Appl. Electron. Mater.* 2020, 2, 3650–3657). Thus, CFO film with different growth orientations own different magnetic easy axis directions and magnetic properties, as demonstrated in *Sci. Rep.* 2016, 6, 30074 and *Appl. Phys. Lett.* 2013, 103, 092405. A typical example is given in **Fig. R10a**, CFO (001) film exhibits a stronger anisotropy between in-plane and out-of-plane directions than CFO (111) (*Appl. Phys. Lett.* 2013, 103, 092405). Besides, the magnetic properties are strongly dependent on

the deposition temperature and substrates (e.g. additional epitaxial strain discussed in our **Reply 2**). A typical example is given in **Fig. R10b**, the as-grown CFO film (with strain) and transferred CFO film (release of strain) exhibit different magnetic anisotropy (*ACS Nano* 2020, 2, 3650–3657).

Fig. R10 | (a) The Figure extracted from *Appl. Phys. Lett.* 2013, 103, 092405, which describes CFO (001) exhibits a stronger anisotropy between in-plane and out-of-plane directions than CFO (111). (b) The Figure extracted from *ACS Nano* 2020, 2, 3650, which describes the as-grown CFO film (with strain) and transferred CFO film (release of strain) exhibit different magnetic anisotropy. (c) The Figure extracted from *ACS Appl. Mater. Interfaces* 2017, 9, 7297, which describes CFO (111) film show no significant difference between out-of-plane and in-plane hysteresis loops. (d) The Figure extracted from *J Mater Sci: Mater Electron* 2017, 28, 446, which describes CFO (111) film deposited at 750°C exhibits a magnetic easy axis along the [111] direction. (e) Out-of-plane and in-plane hysteresis loops of our CFO (111) nanosheet samples deposited at 700°C, which evidences an out-of-plane magnetic easy axis.

As for CFO (111) system, the main research object of our manuscript, (1) Liu et al. demonstrated that CFO (111) film shows no significant difference between out-of-plane and in-plane hysteresis loops (**Fig. R10c**, *ACS Appl. Mater. Interfaces* 2017, 9, 7297). (2) Kumar et al. demonstrated that CFO (111) film deposited at 750°C exhibits a magnetic easy axis along the [111] direction (**Fig. R10d**, *J Mater Sci: Mater Electron*

2017, 28, 446–453). Such magnetic properties are similar to our results. **Fig. R10e** shows the out-of-plane and in-plane hysteresis loops of our CFO nanosheet samples (deposited at 700°C), which evidences an out-of-plane magnetic easy axis (i.e. higher magnetization and larger coercive force).

Based on your comment, the corresponding discussion has been added to the revised manuscript as “*Although CFO with (111) growth orientation exhibits a relatively weak magnetic anisotropy than (100), the comparison between M-H curves under OOP and IP magnetic field at room temperature evidences an OOP magnetic easy axis, in accord with the calculated MAE and previous report⁴³*” (page 12, lines 4-7). The relevant reference has been quoted.

Comment 4: *Figure 3, close observation of MH plots suggest that no considerable change in coercivity and magnetisation for the in-plane and out-of-plane measurements. This happen when the sample is randomly oriented. Thus, these results must be correlated with the crystal orientation. TEM measurement does not support the magnetic properties.*

Our reply: In our **Reply 1**, combining our previous studies and newly-added XRD data, we have demonstrated that CFO nanosheets is not randomly oriented and [111] is the preferred growth orientation. In our **Reply 3**, we demonstrated that the weak magnetic anisotropy between in-plane and out-of-plane directions is the typical characteristic of CFO film with (111) growth orientation. Meanwhile, the corresponding discussion has been added to the revised manuscript. We hope the crystal structure of CFO nanosheet has been more clearly described in this revision.

Comment 5: *Authors mentioned the semiconducting nature of the CFO, however none of the measurement appeared in the manuscript to support the claim. At least resistivity measurement with temperature should have been carried out to support the claim. Hall measurements could also be possible.*

Our reply: We thank the Reviewer to point out this issue. In our previous studies, the semiconducting nature of CFO nanosheets is demonstrated by temperature-dependent

electrical measurement, FET measurement and first principles simulations. Following your critical suggestion, we have performed further temperature-dependent electrical measurements on CFO nanosheets with various thicknesses, and summarized the corresponding temperature-dependent resistance and activation energy (**Fig. R11**). The linear $I_{DS}-V_{DS}$ characteristics indicate the devices have good Ohmic contact, thus the influence of contact can be neglected. As shown in **Fig. R11f**, the device resistance decreases monotonously with the increasing temperature, revealing the semiconducting nature of CFO nanosheets. Hall measurements is not proceeding well due to the highly resistive nature (approaching the detecting limitation of our PPMS) of CFO nanosheets. In the future, we would like to further upgrade our instruments to investigate the Hall Effect of CFO nanosheets.

Based on your comment, we further confirmed the semiconducting nature of CFO nanosheets by temperature-dependent resistance measurement. The corresponding characterization and discussion were added to the revised manuscript and SI. Please see the marked text on page 9 (lines 11-13, 20-23) and Supplementary Fig. 10.

Fig. R11 | (a-e) Temperature-dependent electrical measurements of CFO nanosheet device with various channel thicknesses, as well as the corresponding device image (left) and Arrhenius plot of conductance (inset). (f) Temperature-dependent resistance of CFO nanosheet device with various channel thicknesses. (g) Plot of activation energy (E_a) along with channel thickness.

Response to Reviewer #3

In their manuscript « Ultrathin ferrite nanosheets for room-temperature two-dimensional ferromagnetic semiconductors » Cheng et al. describe the growth and characterization of 2D CFO nanosheets. In particular their magnetic characterizations (transport, VSM, MFM, MOKE) exhibit clear ferromagnetic behavior above room temperature, with long term ambient stability.

This study announced goal is interesting regarding several scientific aspects: (1) it showcases a novel material 1-unit 2D-CFO which is original for the blooming field of 2D ferromagnets, (2) it puts forward both large-scale growth by CVD approach and a large panel of characterizations (chemical, crystallographic, transport, magnetism...) of this novel 2D ferromagnet, (3) it exhibits properties in strong position compared to the state of the art for a 2D material with ferromagnetism recorded at and above room temperature, (4) finally it opens the discussion about 2Ds to novel families of materials (1-unit thick 2D spinel magnetite).

However, there is a strong clarity issue concerning the material which is probed in displayed experiments. This study relates to 1 unit cell CFO as a new material (authors write : “Figure 1f shows the atomic force microscope (AFM) image of a typical CFO nanosheet, which features flat surface and ultrathin thickness of ~1.7 nm corresponding to one unit cell along the [111] direction”) but at several places it is unclear if authors are dealing with regular multi-unit (bulk) CFO (see Zhou Appl. Phys. Lett. 88, 013111 (2006), Dhakal Journal of Applied Physics 107, 053914 (2010)) or with the one-unit material they introduce initially as the focus of the study.

Our reply: First of all, we are grateful to the Reviewer for his/her positive evaluation that highlight the importance and general interest of our work.

We also thank the Reviewer for the instructive comments that stimulate us to further improve our manuscript. In the revised edition, we have addressed the Reviewer's questions and revised our manuscript accordingly. In particular, we have annotated the specific thickness of CFO nanosheet which is probed in displayed experiments.

Before we give our point-by-point responses, we would like to clarify the main research object of our manuscript: that is, CFO nanosheets.

First, it is evident that the materials with decreasing thickness has certainly resulted in unexpected properties, but when exactly a thin crystal should be considered a 2D material may be a matter of debate — or of definition (*Nat. Phys.* 2015, 11, 625–626). It becomes more complicated when it comes to non-layered system. The only specific definition we could find at present is in *Chem. Rev.* 2017, 117, 9, 6225–6331 by Hua Zhang et al., that is, “Ultrathin two-dimensional (2D) nanomaterials represent an emerging class of nanomaterials that possess sheet-like structures with the lateral size larger than 100 nm, or up to a few micrometers and even larger, but the thickness is only single- or few-atoms thick (typically less than 5 nm).” Thus, the term “2D nanosheet” does not specifically refer to “monolayer or 1-unit-cell”, and is normally used to distinguish the individual ultrathin atomic crystal from the regular bulk or film form. Examples include:

(1) In the literature entitled “Two-dimensional halide perovskite lateral epitaxial heterostructures” (*Nature* 2020, 580, 614–620), all perovskite crystals with thickness of 2-20 nm are referred to as 2D perovskites.

(2) In the literature entitled “Phase engineering of Cr₅Te₈ with colossal anomalous Hall effect” (*Nat. Electron.* 2022, 5, 224–232), all Cr₅Te₈ crystals with thickness of 6-20 nm are referred to as 2D Cr₅Te₈.

(3) In the literature entitled “Phase-controllable growth of ultrathin 2D magnetic FeTe crystals” (*Nat. Commun.* 2020, 11, 3729), all FeTe crystals with thickness of 3-30 nm are referred to as 2D FeTe.

Next, we will explain the difference between our CFO nanosheets and the literature reports mentioned by the Reviewer. In the literature *Appl. Phys. Lett.* 2006, 88, 013111, Zhou et al. mainly studied CFO/PZT double-layer thin film prepared by pulsed-laser deposition. The film thickness is ~30 nm and surface root-mean-square roughness is ~3.7 nm, as shown in **Fig. R12a**. It’s apparently not a nanosheet crystal. Another, in the literature *J. Appl. Phys.* 2010, 107, 053914, Dhakal et al. mainly studied 200 nm-thick CFO film, which is apparently not a nanosheet crystal. By contrast, our CFO nanosheet features atomically smooth surface with root-mean-square roughness of ~0.1 nm, which is comparable to that of 2D layered crystals (*Sci. Adv.* 2019, 5, eaau0906).

After carefully considering your comment, we admit that some thickness values of CFO nanosheet which is probed in our experiments are not well clarified. According to the Reviewer’s suggestions, we have conducted additional characterizations on 1-unit-cell CFO nanosheets and changed the statement of some inappropriate “2D CFO” into “CFO nanosheet”. The specific thickness of CFO nanosheet is also annotated.

Now, we believe the reader can know the specific thickness of CFO nanosheet probed in our experiments and properly follow our work.

Fig. R12 | (a) The Figure extracted from *Appl. Phys. Lett.* 2006, 88, 013111, which shows AFM image of CFO/PZT double-layer thin film. With a surface root-mean-square roughness of ~ 3.7 nm, it’s apparently not a nanosheet crystal. (b) AFM image of our CFO nanosheet, which features atomically smooth surface with root-mean-square roughness of ~ 0.1 nm.

Our point-by-point responses are stated as below.

Comment 1: *Why is the cross-section TEM showing multiple unit cells (Fig 1e), while the AFM shows the thickness of only 1 unit cell (Fig 1f)?*

Our reply: We thank the Reviewer to point out this issue. In our previous version, we studied the cross-sectional HAADF-STEM image of a relatively thick CFO nanosheet sample. While the identification of 1-unit-cell cross-section sample, which including three subcells with different atomic distributions, is a real challenge due to: (1) during the cross-section sample preparation, the crystal lattice of 1-unit-cell CFO sample is easily destroyed by the destructive ‘high-energy’ metal deposition and focused ion

beam milling (e.g. atom or cluster bombardment and strong local heating); (2) during the STEM characterizations, 1-unit-cell CFO sample is extremely sensitive to electron beams and experiences knock-on damage and radiolysis, which can survive for only a short time even at low-dose STEM measurements. Besides, the magnetic atoms (Co/Fe) may influence the electron focusing.

Following your critical suggestion, we have carried out a certain amount of study on 1-unit-cell CFO cross-section sample by carefully adjusted the protective layer and measurement conditions, and obtained clearer cross-sectional HAADF-STEM atomic images. As shown in **Fig. R13**, the image shows high crystalline quality in 1-unit-cell-thick CFO nanosheet, which owns a periodic rectangular pattern in the vertical direction, corresponding to three subcells with different atomic distributions. Unexpectedly, the out-of-plane lattice of the bottom subcell expands to ~ 0.54 nm, whereas the out-of-plane lattice of other subcells (~ 0.48 nm) are almost the same as that of the subcells observed in the thick CFO cross-section sample (Fig. 1e). This resembles the ultrathin freestanding BiFeO₃ films (*Nature* 2019, 570, 87-90) and may account for the lost magnetism of 1-unit-cell CFO.

Based on your comment, we studied the cross-sectional HAADF-STEM image of 1-unit-cell CFO and added the corresponding discussion into the revised Manuscript as “*We also performed cross-sectional studies on 1-unit-cell CFO nanosheet (Fig. 1g and Supplementary Fig. 7). Unexpectedly, the out-of-plane lattice of the bottom subcell expands to ~ 0.54 nm, whereas other subcells (~ 0.48 nm) are almost the same as that of the subcells observed in the thick CFO cross-section sample (Fig. 1e). This resembles the ultrathin freestanding bismuth ferrite films and may account for the lost magnetism of 1-unit-cell CFO¹⁶*” (page 7, lines 8-14). The corresponding characterizations were added to Fig. 1g and Supplementary Fig. 7.

Fig. R13 | Cross-sectional HAADF-STEM image of 1-unit-cell CFO nanosheet. (a) SEM image of the 1-unit-cell CFO nanosheet sample prepared by Focused Ion Beam milling. (b,c) Cross-sectional HAADF-STEM image of 1-unit-cell CFO nanosheet, which shows a periodic rectangular pattern in the vertical direction, corresponding to three subcells with different atomic distributions.

Comment 2: *Why transport has been provided on a thick multilayer CFO (figure 2f) while the focus is on the 1 unit material? It is unclear if Figure 2h is on such a multilayer or on 1L CFO as they use the term “2D-CFO” for the thicker sample of Figure 2f. Authors should clarify and restrict the use of “2D CFO” for the 1-unit CFO and “nanosheet CFO” otherwise.*

Our reply: We thank the Reviewer to point out this issue. We have performed more electrical measurements on CFO nanosheets with 1-unit-cell thickness, and summarized the corresponding temperature-dependent resistance and activation energy (**Fig. R14**). The linear $I_{DS}-V_{DS}$ characteristics indicate the devices have good Ohmic contact, thus the influence of contact can be neglected. As shown in **Fig. R14f**, all the device resistance decreases monotonously with the increasing temperature, revealing the semiconducting nature of CFO nanosheets. Besides, the activation energy, defined as the energy difference between Fermi level and mobility edge, increases with the

decreasing thickness. Such trend is consistent with the previously reported 2D van der Waals ferromagnetic device (*Nat. Commun.* 2017, 8, 14410).

Fig. R14 | (a-e) Temperature-dependent electrical measurements of CFO nanosheet device with various channel thicknesses, as well as the corresponding device image (left) and Arrhenius plot of conductance (inset). (f) Temperature-dependent resistance of CFO nanosheet device with various channel thicknesses. (g) Plot of activation energy (E_a) along with channel thickness.

As for vertical device measurements (Figure 2h), we have provided its AFM image and thickness in the Supplementary Fig. 12. As suggested, the specific thickness value of CFO nanosheet is annotated in the caption of Figure 2h (page 8) in the revised Manuscript.

As for the use of “2D CFO”, this issue has been discussed in detail in our above reply. According to the Reviewer’s suggestions, we have changed the statement of some inappropriate “2D CFO” into “CFO nanosheet”. The corresponding specific

thickness of CFO nanosheet is also annotated.

In short, based on your comment, we have performed more electrical measurements on CFO nanosheets with 1-unit-cell thickness, and changed the statement of some inappropriate “2D CFO” into “CFO nanosheet”. The corresponding characterization and discussion were added to the revised manuscript and SI. Please see the marked text on page 9 (lines 20-21) and Supplementary Fig. 10.

Comment 3: *Authors should confirm that the magnetic characterizations on 2D-CFO (fig3) deals with 1 unit cell material. I note as well that the technique for measurement of magnetic properties in Fig 3 is not defined.*

Our reply: We thank the Reviewer to point out this issue. As for the VSM measurement (Fig. 3), it is carried out on an ensemble of CFO nanosheets. Thus the average signal is obtained when using VSM as the measurement method. Following your critical suggestion, we statistically analyzed the thickness and orientation of CFO nanosheets on a large scale. As shown in **Fig. R15**, statistics of AFM images demonstrate that the thickness of CFO nanosheets is mainly distributed in $\sim 2\text{-}4$ nm. Even though it is difficult to synthesize non-layered samples with uniform thickness at present, we obtained a relatively narrow thickness distribution of the resulting nanosheets. Such measurement technique is a common approach in 2D magnetic studies, especially in the case of CVD. Recent examples include: 2D CrTe crystals (*Nat. Commun.* 2021, 12, 5688), 2D CrTe₂ crystals (*Nat. Commun.* 2021, 12, 809), 2D Fe₇Se₈ crystals (*Nano Lett.* 2022, 22, 1242–1250) and 2D Cr_{1.5}Te₂ crystals (*Nano Lett.* 2021, 21, 9517–9525).

In this period, we are trying to perform VSM measurement on 1-unit-cell CFO nanosheet samples with low coverage on mica substrate. However, the magnetic signal of 1-unit-cell CFO is too weak compared with the overwhelmingly larger background signal from the substrate and beyond the resolution of VSM, as indicated in Wei et al 2021 *2D Mater.* 8 012005. To accurately determine the magnetic properties of a CFO nanosheet with well-defined thickness, we conducted MFM and MOKE measurements. These methods can be combined together and compensate each other, characterizing the magnetic properties of CFO nanosheets from different aspects.

Based on your comment, we added the statistical data, as well as the corresponding description to the revised manuscript and SI. Please see the marked text on page 6 (lines 7-7), page 11 (lines 23-25) and Supplementary Fig. 5.

As for the technique for measurement of magnetic properties in Fig 3, the magnetic susceptibility measurements are performed in a physical property measurement system (PPMS, Quantum Design) equipped with VSM utility. These information has been added to the “Methods - Sample characterization” section (page 17) of the revised Manuscript.

Fig. R15 | Histogram statistics of the thickness (a) and orientation (b) of CFO nanosheets with substrate temperature of 700 °C, smooth curve is the Gaussian fit of the thickness distribution. The resulting nanosheets exhibit a relatively narrow thickness distribution (~2-4 nm) and nearly identical crystallographic orientations.

Comment 4: *Fig4 shows MFM characterizations for thick samples only which is surprising. The 1-unit material is not probed there while all the thicker materials are called “2D CFO”.*

Our reply: Following your critical suggestion, we have changed the statement of some inappropriate “2D CFO” into “CFO nanosheet”. The corresponding specific thickness of CFO nanosheet is also annotated. We agree with the Reviewer that exploring the magnetic properties of 1-unit cell CFO is worthy of researching. In our previous version, we have demonstrated that 2-unit-cell CFO is indeed a hard magnetic material. However, it is quite challenging to accurately detect the weak magnetic signal of 1-unit-cell CFO, since it required highly sensitive instruments and minimal electromagnetic

interference. Based on the multiple testing, more room-temperature MOKE and MFM data are provided (**Fig. R16**).

When the CFO thickness reduced to ~ 2.5 nm, the MOKE signal is no longer a square hysteresis loop despite the obvious remanence at zero magnetic field (a sign of ferro-/ferrimagnetism). Further, no obvious hysteresis loops are observed in sub-2-nm-thick CFO nanosheets. Even so, a sizeable Kerr rotation is observed as a function of the applied magnetic field, suggesting a detected out-of-plane component of magnetization. These results are consistent with MFM tests (**Fig. R16c**). MFM test fails to demonstrate the magnetism of 1-unit-cell CFO. In contrast, the magnetic signal obtained from inner triangle island (corresponding to ~ 2 unit cells) is much stronger.

By combining MFM and MOKE measurements, we can conclude that 2-unit-cell CFO is indeed a hard magnetic material. While for sub-2-nm-thick samples (~ 1 unit cell), we can only clarify that they own a detected out-of-plane component of the magnetization. The reasons could be:

- (1) Enhanced spin fluctuation in reduced dimensions. As compared to the bulk system, the decreased size of ferro-/ferrimagnetic material to a critical dimension may leads to single domain structure. Single domain structures exhibit large magnetic moment. Below critical dimensions, the formation of further domains becomes energetically unfavorable, thus the magnetic moments flip randomly with time and ferro-/ferrimagnetic material behaves as super-paramagnetic (SPM) by losing their magnetic order when the magnetic anisotropy energy becomes comparable to thermal energy. Such behavior is also observed in graphene on antiferromagnet CrSe (*Nat. Electron.* 2020, 3, 604–611, please see **Fig. R16b**), as well as CoFe₂O₄ nanoparticles with an average particle size of 15 nm (*Physica B* 2019, 567, 87-94).

Fig. R16 | (a) Room-temperature MOKE hysteresis loops of ultrathin CFO nanosheets with variable thicknesses. (b) The Figure extracted from *Nat. Electron.* 2020, 3, 604–611, which describes MOKE measurements of graphene on CrSe and a control sample. Kerr rotation of graphene on CrSe shows the magnetized graphene at 12 K and no magnetic signal at room temperature. (c) MFM phase images and AFM images of CFO nanosheets with variable thicknesses measured at room temperature. Scale bars: 3 μm .

- (2) Limited sensitivity of present instruments. Due to the limited sensitivity of MFM and MOKE, it is difficult to determine exactly the weak stray magnetic field emanating from the 1-unit-cell sample, just like many monolayer or 1-unit-cell cases in 2D atomic crystals (*Nature* 2017, **546**, 265–269; *Nat. Commun.* 2021,

12, 5688). Recently, scanning magnetometry based on a single electron spin of a nitrogen-vacancy center in diamond has shown a high sensitivity in probing 2D magnets (*Science* 2021, 374, 1140–1144). Unfortunately, due to the limited condition in our lab and the influence of COVID-19, this characterization couldn't be conducted. In the future, we will conduct more experiments to validate the explanation.

Based on your comment, we have added the corresponding discussion into the revised Manuscript as:

- (a) *“MFM test fails to demonstrate the ferrimagnetism of 1-unit-cell CFO. In contrast, the magnetic signal obtained from inner triangle island (~2 unit cells) is much stronger (Supplementary Fig. 18b).”* (page 13, lines 21-23)
- (b) *“When the CFO thickness reduced to ~2.5 nm, the MOKE signal is no longer a square hysteresis loop despite the obvious remanence (a sign of ferro-/ferrimagnetism). Further, no hysteresis loops are observed in sub-2-nm-thick CFO nanosheets. Even so, a sizeable Kerr rotation is observed as a function of the applied magnetic field, suggesting a detected OOP component of the magnetization. The reasons could be: (a) Enhanced spin fluctuation in reduced dimensions. Below critical dimensions, the formation of further domains becomes energetically unfavorable, thus the magnetic moments flip randomly with time and 1-unit-cell CFO behaves as super-paramagnetic by losing their magnetic order when the magnetic anisotropy energy becomes comparable to thermal energy. Such behavior is also observed in magnetized graphene and CFO nanoparticles^{45,46}. (b) Due to the limited sensitivity of MFM and MOKE, it is difficult to determine exactly the weak stray magnetic field emanating from the 1-unit-cell CFO, just like many monolayer or 1-unit-cell cases in 2D crystals^{7, 14}.”* (page 14, lines 9-21)

Comment 5: *Supplementary materials figures S15 and S16 seem to study materials above 2 unit cells thicknesses. And critically, it seems that the 2 unit-cell layer is not a room temperature ferromagnet, not even ferromagnetic at 80K! This is quite distressing*

regarding the unprobed but claimed 1L CFO room temperature FM!

Our reply: Following your critical suggestion, we have further discussed the magnetic properties of CFO nanosheets in our **Reply 4**: that is, 2-unit-cell CFO is indeed a hard magnetic material at room temperature and 1-unit-cell CFO own a detected out-of-plane component of the magnetization. Besides, we would like to clarify that we didn't claim 1L CFO is room temperature FM in our previous version. Such a misunderstanding may be due to the fact that there is no consensus definition for "2D" in the present academe. To avoid possible misunderstanding, we have changed the statement of some inappropriate "2D CFO" into "CFO nanosheet". The corresponding specific thickness of CFO nanosheet is also annotated.

If authors can provide a convincing clarification on what is the material probed in the different experiments, and particularly justify clearly that the 1-unit cell CFO is indeed a 2D ferromagnet, it would be then possible to assess the supposed novelty.

Our reply: We would like to thank the Reviewer for the thoughtful scrutiny of our manuscript. Based on your suggestions, we have conducted additional characterizations on 1-unit-cell CFO nanosheets, changed the statement of some inappropriate "2D CFO" into "CFO nanosheet", and clarified the specific thickness of CFO nanosheet probed in the different experiments. Now, we believe the reader can properly follow our work. Specifically, we have made the following revisions:

- (1) We studied the cross-sectional HAADF-STEM image of 1-unit-cell CFO.
- (2) We performed more electrical measurements on 1-unit-cell CFO.
- (3) We clarified the test object and technique for magnetic susceptibility (Fig. 3).
- (4) We concluded that 2-unit-cell CFO is indeed a hard magnetic material and 1-unit-cell CFO own a detected out-of-plane component of the magnetization.

We think the characterization is more comprehensive and the discussion is deeper now, and the quality of this manuscript has a great improvement. We hope it can address the Reviewer's concerns and reach the high request of *Nature Communications* now.

Reviewers' Comments:

Reviewer #1:

Remarks to the Author:

The authors have made significant improvements to the manuscript and have clearly addressed all of the reviewer's comments and concerns. I recommend publication as-is in Nature Communications.

Reviewer #2:

Remarks to the Author:

Manuscript is satisfactorily revised and thus it can be published in its present form.

Reviewer #3:

Remarks to the Author:

Dear Editor,

Authors made a substantial revision, which addresses convincingly the raised points. They have defended the interest of studying thin CFO nanosheets beyond the 1-unit cell material with great clarity. The thickness ambiguity issue is now much clarified throughout the manuscript. I am now convinced of the interest of the presented data and recommend publication.

Response to Reviewer #1

The authors have made significant improvements to the manuscript and have clearly addressed all of the reviewer's comments and concerns. I recommend publication as-is in Nature Communications.

Our reply: We would like to thank you for reviewing our paper, we appreciate your insightful comments on our research.

Response to Reviewer #2

Manuscript is satisfactorily revised and thus it can be published in its present form.

Our reply: We would like to thank you for reviewing our paper, we appreciate your insightful comments on our research.

Response to Reviewer #3

Dear Editor,

Authors made a substantial revision, which addresses convincingly the raised points. They have defended the interest of studying thin CFO nanosheets beyond the 1-unit cell material with great clarity. The thickness ambiguity issue is now much clarified throughout the manuscript. I am now convinced of the interest of the presented data and recommend publication.

Our reply: We would like to thank you for reviewing our paper, we appreciate your insightful comments on our research.